# Statistical considerations of nonrandom treatment applications reveal region-wide benefits of widespread post-fire restoration action

Allison B. Simler-Williamson [1,2] & Matthew J. Germino[2]

Accurate predictions of ecological restoration outcomes are needed across the increasingly large landscapes requiring treatment following disturbances. However, observational studies often fail to account for nonrandom treatment application, which can result in invalid inference. Examining a spatiotemporally extensive management treatment involving post-fire seeding of declining sagebrush shrubs across semiarid areas of the western USA over two decades, we quantify drivers and consequences of selection biases in restoration using remotely sensed data. From following more than 1,500 wildfires, we find treatments were disproportionately applied in more stressful, degraded ecological conditions. Failure to incorporate unmeasured drivers of treatment allocation led to the conclusion that costly, widespread seedings were unsuccessful; however, after considering sources of bias, restoration positively affected sagebrush recovery. Treatment effects varied with climate, indicating prioritization criteria for interventions. Our findings revise the perspective that post-fire sagebrush seedings have been broadly unsuccessful and demonstrate how selection biases can pose substantive inferential hazards in observational studies of restoration efficacy and the development of restoration theory.

---

[1] Department of Biological Sciences, Boise State University, 1910 W University Dr, Boise, ID 83725, USA. [2] U.S. Geological Survey, Forest and Rangeland Ecosystem Science Center, 230 N. Collins Rd., Boise, ID 83702, USA. ✉email: allisonsimlerwil@boisestate.edu

Restoration is an essential tool for counteracting losses in habitat, biodiversity, and other ecological attributes that are threatened by global change drivers, such as changing disturbance regimes and species invasions. However, outcomes of restoration treatments can be highly variable–even across similar sites and treatments[1,2]–and this variability in both space and time[3,4] can obscure broad efforts to quantify average restoration effectiveness. The lack of predictive power resulting from this variability has inhibited the development of theoretical frameworks in restoration ecology[2]. Given escalating local and global stressors and the limited resources available to do restoration, there is an urgent need to be able to predict more broadly which treatments will be successful, as well as understand where and when restoration results in the greatest ecological gains[5].

Randomized, manipulative experiments, in which treatments are applied and assessed against comparable control plots, are frequently used to study restoration efficacy. These efforts are costly and typically do not match the spatiotemporal scope of actual management efforts, which can occur over 10's to 1000's of hectares or across decades[6,7]. Recent advances in the remote sensing of certain plant types have allowed for estimation of population recovery across broader spatial and temporal extents, presenting an opportunity to explore restoration outcomes at a larger scale[8–12]. However, the inferences from this approach have a contrasting limitation: conservation actions, including restoration treatments, are rarely randomly distributed, due to variation in ecological value, conservation need, social willingness, costs, and institutional resources across heterogenous landscapes[13–16].

This nonrandom, often strategic, deployment of restoration treatments can lead to statistical bias (referred to as "selection bias") and spurious inference about the effectiveness of management actions when classic regression approaches are used[17–20]. A key assumption of these statistical approaches is that no confounding, unobserved factors are correlated both with the response and the treatment effect of interest[20]. Thus, as large-scale remotely sensed products and aggregated datasets become more readily available, selection biases have the potential to hinder inference about restoration successes or failures from these observational datasets. Recent meta-analyses have explored the efficacy of several of the world's most common restoration efforts, including soil recovery following agricultural conversion[21], active forest regeneration following deforestation[22], or the control of invasive plants[23]. Notably, many assessments of restoration have failed to observe widespread benefits or full ecosystem recovery associated with treatments, a pattern potentially attributable to the inferential hurdles faced by observational analyses[14,24].

Statistical approaches can minimize the effects of selection biases when inferring treatment effects from large, observational datasets[20]. Researchers may include additional covariates in regression analyses to consider possible confounding factors, or similarly, "propensity score matching" accounts for observed drivers of selection bias by matching observations with similar probabilities of treatment and different treatment statuses based on a set of measured covariates. However, each of these approaches assume that all factors influencing treatment allocation have been observed[25]. When repeated observations from before and after the treatment are available (such as in Before-After Control-Impact designs (BACI)), "difference-in-differences" estimation can leverage repeated observations to account for time-invariant measured and unmeasured sources of bias[26]. Difference-in-differences (DiD) estimation identifies: (a) the persistent differences between treated and untreated groups (which may arise from nonrandom deployment of treatments), (b) the expected changes in these groups over time (under the assumption that groups would have had parallel trends, in the absence of treatment), and (c) the treatment effect, identified as the average observation's departure from its expected temporal trajectory when treatment is applied (identified using an interaction term between an observation's group and time period). Within-estimator panel regression extends the DiD regression structure to consider additional sources of bias, such as numerous post-treatment time points, multiple groups, or time-varying sources of heterogeneity included as covariates. Though these before-after control-impact designs can be employed in small-scale assessments of treatment efficacy, repeated measurements are often costly, impractical, or impossible to collect (i.e. due to unpredictable locations of ecological impacts)[19]. Thus, only a small proportion of ecological impact studies, including assessments of restoration efficacy, typically apply these approaches[27,28].

We examined how statistical approaches that differentially address selection biases influence our estimation of the efficacy of one of the most widely deployed restoration treatments globally––post-fire seeding of fire-intolerant shrubs. Sagebrush-steppe ecosystems once covered 620,000 km$^2$ of western North America[29] but are now one of the continent's most threatened vegetation types. The foundational species of these habitats, sagebrushes (*Artemisia spp.*, especially big sagebrush, *A. tridentata*), have been challenged by climate change, conifer encroachment, and the invasion of an invasive annual grass, *Bromus tectorum* or 'cheatgrass', which has contributed to more frequent and larger wildfire occurrence in this region[30,31]. Altered fire regimes have locally eradicated fire-sensitive sagebrush, as most sagebrushes do not resprout, have short seed longevities (<~2 years), and disperse over short distances (<~2 m)[32]. This has led to propagule limitation in increasingly large burned patches and necessitated widespread investments in restoration seeding treatments (dispersed both aerially and on the ground) to facilitate post-fire regeneration[6].

Since 1990, the Bureau of Land Management (BLM), which manages more than half of the land area within the Great Basin, has invested more than $100 million in sagebrush seed for restoration actions, impacting more than 6000 km$^2$ of the Great Basin over the last century[6]. Yet, like many other widely deployed restoration treatments, evidence for the broad "success" of these treatments is variable from regionwide observational studies[33–37] and from investigations targeting specific ecoregions or fires[38–42]. Whether these seeding treatments have been broadly successful and how effectiveness can be quantified has been a long-standing administrative question[43]. An analysis based on field observations of nearly 100 historic post-fire sagebrush seedings and paired unseeded sites concluded that seeding was ineffective[34]. A larger-scale observational field study found that probability of *A. tridentata* occurrence among treated areas increased with repeated seeding actions, although the study did not compare impacts in seeded and unseeded locations[35]. Other research using region-wide, plot-based vegetation cover data found positive correlations between a variety of restoration actions (including post-fire seeding, as well as other treatments) and *Artemisia spp.* occurrence (but not abundance), although the authors cautioned that possible differences in the biophysical characteristics of treated and untreated sites had not been considered in the analysis[44].

Recent restoration planning frameworks provide general recommendations for the prioritization of treatment sites based upon their resistance to cheatgrass invasion, their resilience to wildfire events, and their value as sage-grouse habitat[5,45–47]. However, due to the unpredictable nature of wildfire events, interannual climatic variation, the potentially stochastic availability of economic and biological resources, and heterogeneity in regional and local interpretation of frameworks, the extent to which on-the-ground restoration implementation has historically aligned with these conceptual guidelines remains unclear.

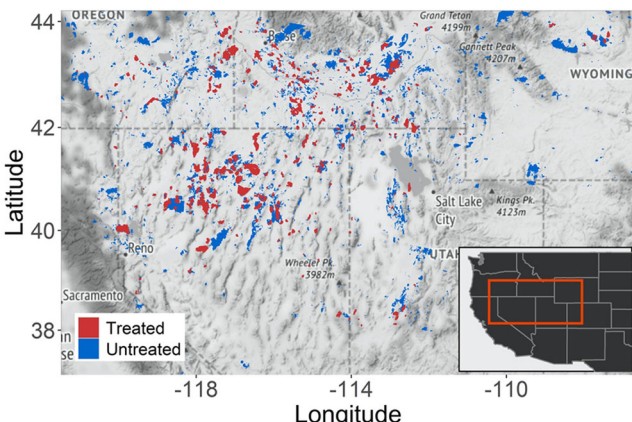

**Fig. 1 Study area description.** Burned areas that received *Artemisia* seeding treatments (as recorded by the Land Treatment Digital Library) and or remained untreated following fires occurring between 1986 and 2001 in the western United States. Areas that burned twice during this period were excluded from the analysis (Map data ©2021 Google).

In this work, we quantify how observed and unobserved sources of selection bias (emerging from the nonrandom deployment of treatments) affect widespread inferences of restoration "success" by examining post-fire sagebrush seedings following more than 1500 wildfires occurring over a 20-year period between 1985 and 2005 across the western U.S. (Fig. 1). Specifically, we examine the following questions: (1) Which biophysical characteristics differ between treated and untreated burned areas?; (2) How does the statistical consideration of the nonrandom application of restoration influence the size and direction of identified treatment effects?; (3) Are post-fire seeding treatments improving sagebrush recovery across a broad spatio-temporal extent, and do effect sizes vary across the same bio-physical characteristics determining treatment implementation?

We integrated the Land Treatment Digital Library (LTDL), a long-term catalog of more than 50,000 management actions on BLM land across the western U.S.[6,48], with estimates of sagebrush cover in 20,000 treated and untreated burned locations derived from satellite imagery, using the Rangeland Condition Monitoring Assessment and Projection (RCMAP, formerly known as the National Land Cover Database's "Back in Time" Sagebrush Rangeland Fractional Component[8,11]). We then examined the biophysical correlates of treatment deployment and treatment effect sizes identified by three comparative approaches, relative to a null "naïve" model in which neither observed nor unobserved sources of selection bias are considered (Table 1). We compared: (1) regression models that considered observed sources of selection bias (such as climate variables and other measured biophysical characteristics of sites that have been commonly integrated into previous studies in this system), either by conducting propensity score matching before analysis or by incorporating environmental covariates directly; (2) DiD estimation, which leverages pre- and post-treatment measures to account for unobserved, time-invariant differences between treated and untreated sites; and (3) within-estimator panel regression, which generalizes DiD estimation to incorporate unobserved heterogeneity associated with multiple time periods or measured time-varying confounders.

Our results provide evidence that widespread post-fire sagebrush seeding efforts may have been more successful in increasing plant cover than previously quantified over broad spatial extents, and that there is significant variation in treatment efficacy across climatic gradients relevant to the global change ecology of this foundational dryland shrub. Invasive annual grasses are expected to further spread across western North America, and their roles as wildfire fuels, in addition to changing climatic conditions, will continue to accelerate fire frequencies and expand the total area requiring restoration intervention[49,50]. Remotely sensed data products and other large-scale observational datasets that monitor plant populations are increasingly accessible. Thus, explicit consideration of nonrandom application of treatments, including the ecological, social, and economic factors shaping these decisions, is essential to advancing restoration theory over broad spatiotemporal extents and prioritizing treatment sites, given limited management resources and accelerating anthropogenic threats.

## Results

**Sources of selection bias in restoration treatments**. In the dataset of randomly selected burned sagebrush habitat locations, treated and untreated sites systematically differed in their bio-physical properties, indicating substantial potential for selection bias in the assessment of restoration treatment effects using observational datasets, such as RCMAP and LTDL (Fig. 2A, B, Supplementary Figs. 1, 2). Within ecoregions where restoration on BLM land commonly occurred, managers tended to conduct post-fire seedings in areas that had low-to-moderate amounts of sagebrush before fire (treatment probability maximized at 12.5% cover; Fig. 3A) and in areas with less surviving, post-fire sagebrush cover (Fig. 3B). Burned sites with greater than 13.3% surviving sagebrush cover had less than a 50% chance of receiving treatment. The probability of burned BLM lands receiving seeding was also greatest in areas with relatively dry, warm springs, compared to the broader climatic range of sagebrush ecosystems (maximized where mean November-April total precipitation was 459 mm and mean spring temperature was 6.01 °C; Fig. 3C, D, respectively). Areas with mean spring temperatures less than 0.60 °C were more likely to remain untreated than receive seeding. Probability of receiving treatment also increased at sites with soils containing a greater proportion of fine clay particles (Fig. 3F).

The probability of areas receiving seeding also decreased in larger burned areas (Fig. 2B) and was greatest at intermediate distances from major roads (Figs. 2B, 3E). Recorded seeding efforts were not equally distributed among ecoregions, and instead were more likely to occur in the Snake River Plain, Idaho Batholith, and Northern Basin, Wyoming Basin, and Central Basin and Range ecoregions (Fig. 2B).

The matching process, which incorporated covariates included in past prioritization frameworks and studies of sagebrush restoration[34,35,51,52], resulted in a subset of the data in which seeded and unseeded locations were similar in terms of these biophysical characteristics (Fig. 2B). Sites also had similar overall probabilities of receiving restoration, despite having different treatment statuses (Fig. 2A), using a caliper size equivalent to 20% of the standard deviation of the mean propensity score. Following matching, the effects of each of the included covariates in the propensity score model did not differ from zero (Fig. 2B), and the means for each covariate did not statistically differ between treated and untreated groups (Supplementary Figs. 1, 2). The matching process eliminated observations in most montane, foothills, and Great Plains systems containing >0% estimated sagebrush cover, where post-fire restoration treatments were relatively rare.

**Variation in estimated treatment effects with approach**. Estimates of restoration efficacy varied considerably depending on the statistical approach used and how measured and unmeasured sources of selection bias and time-invariant and time-varying

**Table 1 Summary of model structures used in comparative analysis of treatment effectiveness.**

| Sources of bias considered | Modeling approach | Model Structure | Sample size |
|---|---|---|---|
| None | 1. "Naïve" model (Difference in means) | $y_i \sim$ Negative binomial$(\mu_i, \theta)$ (1a) <br> $\log(\mu_i) = \alpha + \beta(\text{group}_i) + \varepsilon_i$ (1b) | $n = 20{,}000$ observations of sagebrush cover 10 years post-fire, in 1539 fires |
| Selection bias associated with measured, time-invariant site characteristics | 2. Regression following propensity score matching | $y_i \sim$ Negative binomial$(\mu_i, \theta)$ (2a) <br> $\log(\mu_i) = \alpha + \beta(\text{group}_i) + \varepsilon_i$ (2b) | $n = 11{,}012$ "matched" observations of sagebrush cover 10 years post-fire, in 940 fires |
| | 3. Regression with environmental covariates (with varying intercept for fire identity) | $y_{ik} \sim$ Negative binomial$(\mu_{ik}, \theta)$ (3a) <br> $\log(\mu_{ik}) = \alpha_k + \beta(\text{group}_i) + \boldsymbol{\omega}(\text{X}) + \varepsilon_{ik}$ (3b) <br> $\alpha_k \sim$ Normal$(\alpha, \sigma)$ (3c) | $n = 20{,}000$ observations of sagebrush cover 10 years post-fire, in 1539 fires |
| Selection bias associated with unmeasured time-invariant group characteristics | 4. Difference-in-differences regression model (with varying intercepts for location and fire identity) | $y_{ijk} \sim$ Negative binomial$(\mu_{ijk}, \theta)$ (4a) <br> $\log(\mu_{ijk}) = \alpha_{jk} + \tau(\text{time}_{ijk}) + \gamma(\text{group}_{ijk}) + \beta(\text{time}_{ijk} * \text{group}_{ijk}) + \varepsilon_{ijk}$ (4b) <br> $\alpha_{jk} \sim$ Normal$(\alpha_k, \sigma_j)$ (4c) <br> $\alpha_k \sim$ Normal$(\alpha, \sigma_k)$ (4d) | $n = 40{,}000$ observations pre-treatment (year 0 post-fire) and 10 years following treatment, in 1539 fires |
| Selection bias associated with: 1) unobserved characteristics of timepoints and groups; and 2) measured time-varying and group-varying factors (e.g. weather). | 5. Within-estimator panel regression model (with varying intercepts for location and fire identity) | $y_{ijk} \sim$ Negative binomial$(\mu_{ijk}, \theta)$ (5a) <br> $\log(\mu_{ijk}) = \alpha_{jk} + \tau(\text{time since treatment}_{ijk}) + \gamma(\text{group}_{ijk})$ <br> $+ \beta(\text{treatment indicator}_{ijk})$ <br> $+ \boldsymbol{\omega}(\mathbf{W}) + \varepsilon_{ijk}$ (5b) <br> $\alpha_{jk} \sim$ Normal$(\alpha_k, \sigma_j)$ (5c) <br> $\alpha_k \sim$ Normal$(\alpha, \sigma_k)$ (5d) | $n = 220{,}000$ observations from year 0 (pre-treatment) to year 10 following treatment, in 1539 fires |

*y* represents individual observations (*i*) of sagebrush percent cover, which are nested within locations (*j*, where repeated measures are used) and fires (*k*). *μ* represents the expectation for *y*, *α* represents global intercepts (with varying intercept components $\alpha_j$ and $\alpha_k$ for location and fire identity, which are normally distributed with standard deviations (*σ*) associated with each). Group is a categorical variable indicating whether an observation was in treated or untreated groups. Time indicates whether the observation is pre-treatment (year 0 postfire) or post-treatment application (year 10 postfire) in DiD models. In the within-estimator panel regression model, time since treatment is a categorical variable for the observed timepoint (0–10 years post-treatment) and treatment indicator represents whether the treatment has occurred at a site by the observed timepoint. **X** and **W** represent matrices of either time-invariant biophysical covariates or time-varying weather variables (described in the text) with an associated vector of parameters **ω**. Time-invariant biophysical characteristics were selected based on their inclusion in frameworks for prioritization of sagebrush restoration sites or in past studies of sagebrush recovery as "control" variables. In all models, the parameter associated with treatment application is indicated by *β*.

confounders were integrated (model structures are summarized in Table 1). The "naïve" null model, which did not account for sources of selection bias and contained only a variable for treatment status, indicated that treated sites had, on average, −0.8% less sagebrush cover 10 years following post-fire reseeding efforts, compared to untreated sites within sagebrush habitat (Fig. 4).

In turn, approaches that incorporated only observed environmental sources of selection bias suggested that post-treatment sagebrush cover did not clearly differ between treated and untreated sites (Fig. 4). An analysis of the subset of the observations that remained following the propensity score matching process (which balanced the systematic differences between sites for several biophysical variables; Fig. 2B, Supplementary Figs. 1, 2) resulted in a neutral treatment effect (75.7% probability that the treatment effect ≥0). A regression approach that incorporated a series of environmental covariates that have been accounted for in past studies of sagebrush recovery (as well as a varying intercept for the identity of the fire that impacted each site) estimated a weakly positive mean effect of treatment (+0.26% sagebrush cover, with 95% credible intervals containing zero).

However, using DiD and within-estimator panel regression, we detected positive impacts of seeding efforts on restoration outcomes across the spatiotemporal extent examined in this study (Fig. 4). Based on DiD estimation, which may account for time-invariant, unobserved differences between treatment groups, sagebrush cover increased by an average of 5.7% by 10 years post-treatment on BLM lands where seeding occurred, compared to the expected levels of recovery if these locations had not received

restoration. The within-estimator panel regression, which additionally considered unobserved heterogeneity specific to multiple post-treatment time points, predicted an increase in sagebrush cover at treated sites of 4.1% by 10 years post-treatment. The within-estimator panel model also included time-varying weather variables for February-April total precipitation and mean temperature, which had small positive effects on sagebrush growth (Supplementary Fig. 6); though, correlation with lagged effects of past years' precipitation and temperature variables could bias direct interpretation of the weather covariates included in this model[26]. Within-sample mean absolute error for DiD and within-estimator panel regressions were 2.85 and 2.50, respectively.

Pre-fire estimates of sagebrush cover for treated and untreated pixels exhibited parallel trends (Supplementary Fig. 3), a key assumption for conducting DiD estimation. Many analyses conduct propensity score matching, based on time-invariant site characteristics, prior to DiD estimation to ensure groups will fulfill the assumption of parallel trends, but we did not find this substantially altered the effect size associated with the treatment parameter in our DiD model (Supplementary Information Fig. 12). Cluster-robust standard errors are commonly used as a post-estimation approach to account for grouping structures among observations in econometrics analyses[53]. Instead, we included varying intercepts in the DiD and within-estimator regression models to account for clustering of observations by location and fire identity within the model's structure[54]. The conclusions drawn from the multilevel models' credible intervals did not strongly differ from inference from frequentist confidence

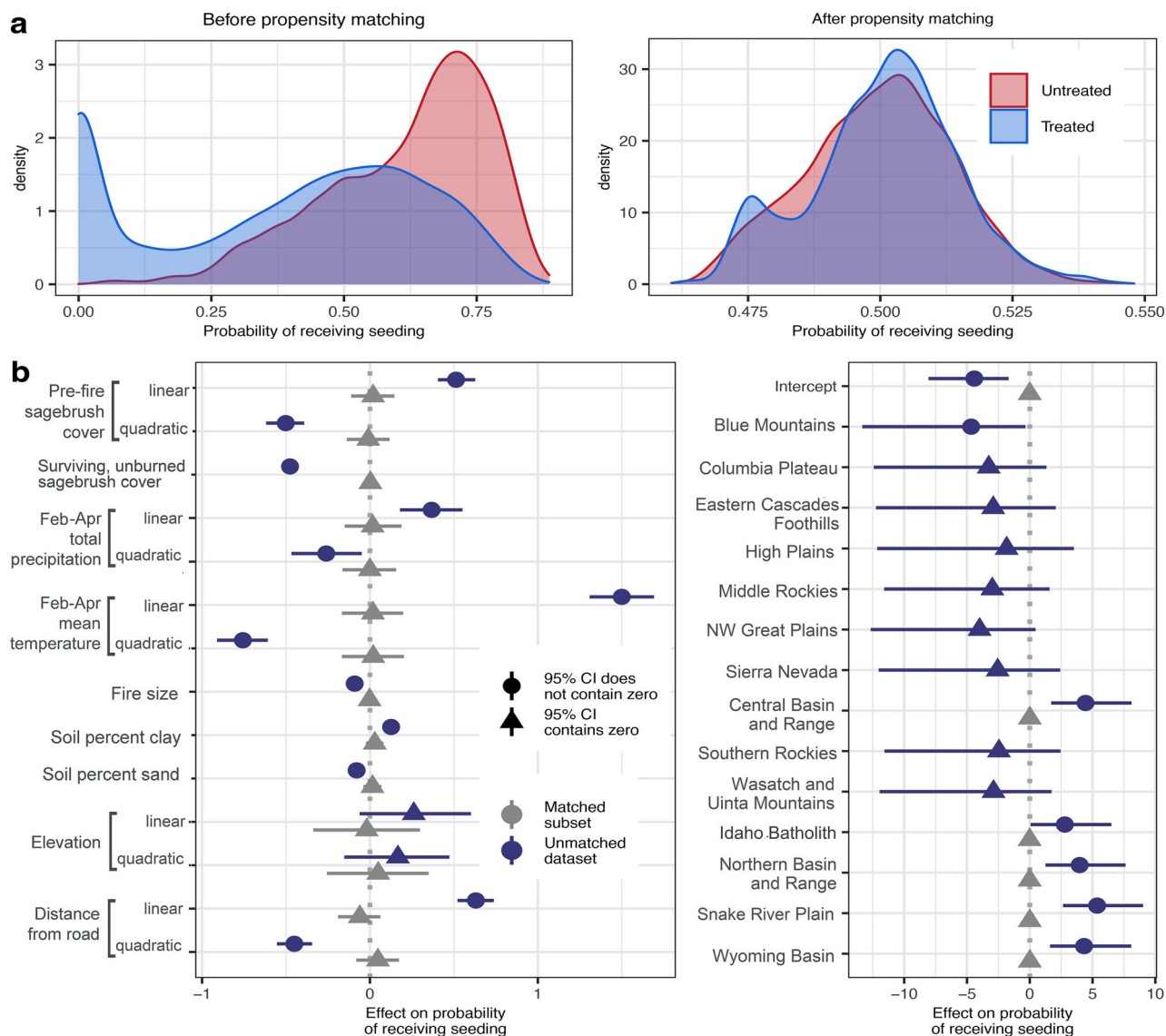

**Fig. 2 Differences in treated and untreated site characteristics before and after propensity score matching. a** The distribution of propensity scores (the probability of receiving seeding treatment) in treated and untreated locations before ($n = 20,000$ locations) and after the matching process ($n = 11,012$ locations); (**b**) Posterior parameter estimates for the effects of environmental covariates in unmatched and matched datasets on the probability of receiving seeding treatments ($n = 20,000$ locations). Symbols represent median parameter estimates and lines represent the 95% credible intervals (CIs) for the parameter estimate, with triangles and dots indicating where 95% CIs included 0 or did not include 0, respectively. In the matched dataset, the effects for some ecoregions are not shown, in cases where the matching algorithm eliminated observations from these ecoregions entirely (i.e. no sufficiently similar pairs of treated and untreated pixels were contained within these ecoregion categories).

intervals calculated using cluster-robust standard errors (Supplementary Fig. 11, Supplementary Tables 2, 3). Standard deviations for varying intercepts in each model are included in the supplementary information (Supplementary Table 1).

**Variation in treatment effects along climatic gradients**. Gains in sagebrush recovery associated with restoration seeding varied substantially along biophysical gradients, indicated by interaction terms between the treatment effect and environmental variables (Fig. 5; Supplementary Fig. 7). Treatment effect size, using DiD estimation, was maximized at intermediate precipitation and cooler temperature values. For instance, sites within the Central Basin and Range ecosystem with an average of 726 mm of November–April precipitation were predicted to have 11.9% more sagebrush cover following post-fire reseeding, compared to gains of 2.4% at the driest locations (Fig. 5B). Seeding was

predicted to increase sagebrush cover by 8.3% at locations with mean February-April temperatures of −5.6 °C but only increased cover by 2.7% at the warmest sites (Fig. 5D). Gains associated with seeding were also maximized at intermediate soil clay proportions (Supplementary Fig. 7). While restoration was commonly applied in sites expected to experience large gains in sagebrush cover from seeding, significant efforts were also allocated to the hottest and driest sites, where treatment effects were predicted to be smallest (Fig. 5B, D).

**Discussion**

Changing fire frequencies, nonnative species invasions, and climate change have caused the decline of myriad foundational plant species globally, including sagebrush species in the genus *Artemisia*[30], prompting a need for improved predictive models of restoration success across large landscapes. Based upon statistical

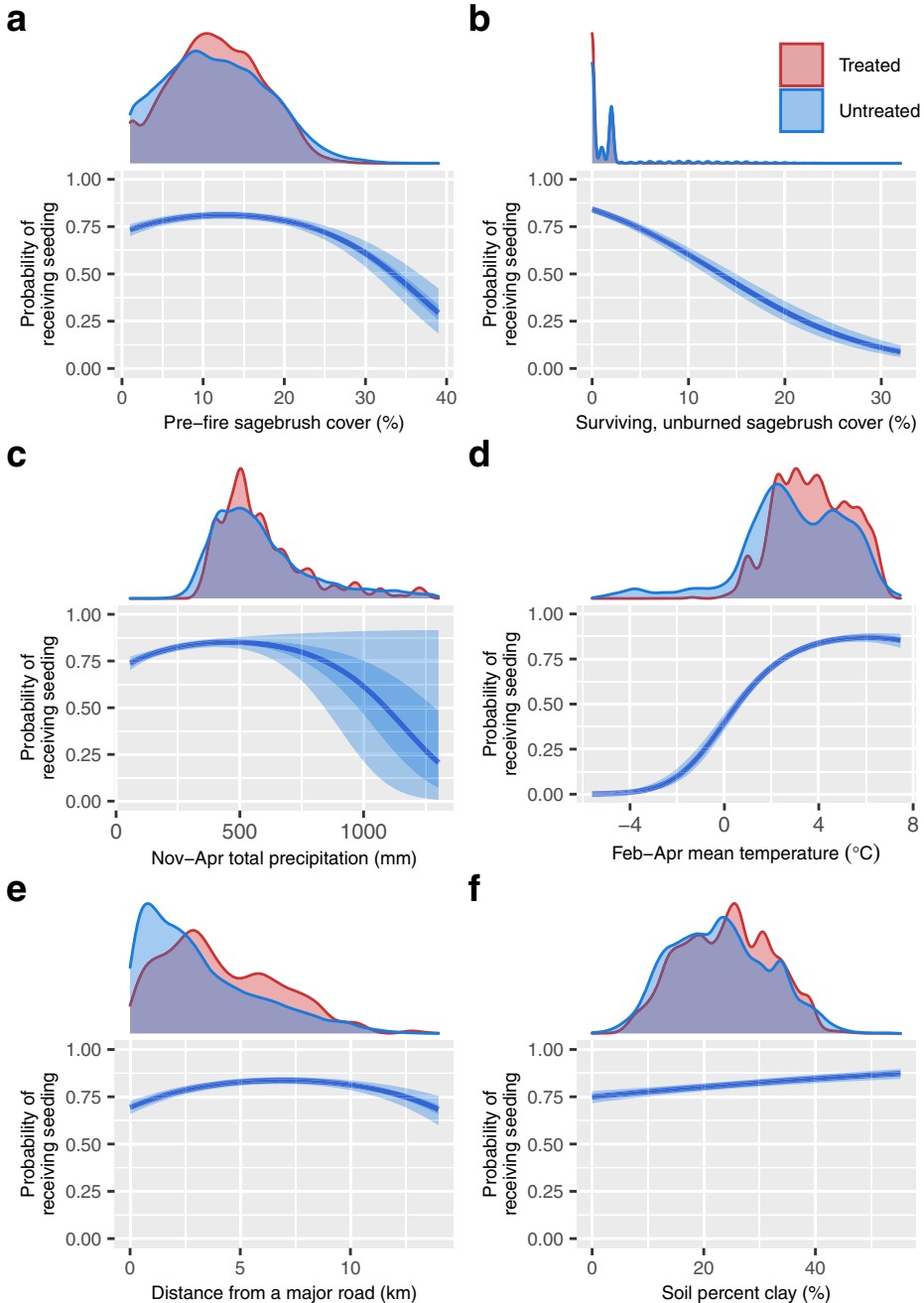

**Fig. 3 Correlates of post-fire seeding treatment application.** Marginal effects of covariates correlated with the occurrence of restoration treatments, illustrating observed sources of selection bias ($n = 20{,}000$ observations). The distribution of observed values for these covariates in treated and untreated sites is shown above each marginal effect panel. Panels illustrate predicted effects of pre-fire sagebrush cover (**a**), surviving, unburned sagebrush cover estimated immediately following the fire (**b**), November-April total precipitation (**c**), February-April mean temperature (**d**), distance from a road (**e**), and soil percent clay (**f**) on treatment probability, while holding all other covariates at their means and assuming observations occurred in a commonly treated ecoregion (Snake River Plain). Solid lines represent median posterior predictions (based on model parameters shown in Fig. 2), with shaded bands indicating 50% and 95% credible intervals around these predictions.

approaches that account for sources of observed and unobserved bias, our results indicate that widespread investment in post-fire sagebrush seeding efforts may be more effective over larger areas or timeframes than previous regionwide studies have suggested (Fig. 4)[33,34,37]. Further, nonrandom treatment locations may have substantially affected past inference for sagebrush seeding treatment effects. Estimates of post-fire restoration treatment effects shifted from negative to neutral to positive with the incremental

consideration of observed and unobserved biases in treatment locations and across time (incorporated into analyses through propensity score matching and environmental covariates, DiD estimation, and within-estimator panel regression, respectively), with significant variation in efficacy across climatic gradients (Figs. 4, 5).

Our results also explicitly quantify the environmental correlates of restoration implementation in a widespread dryland

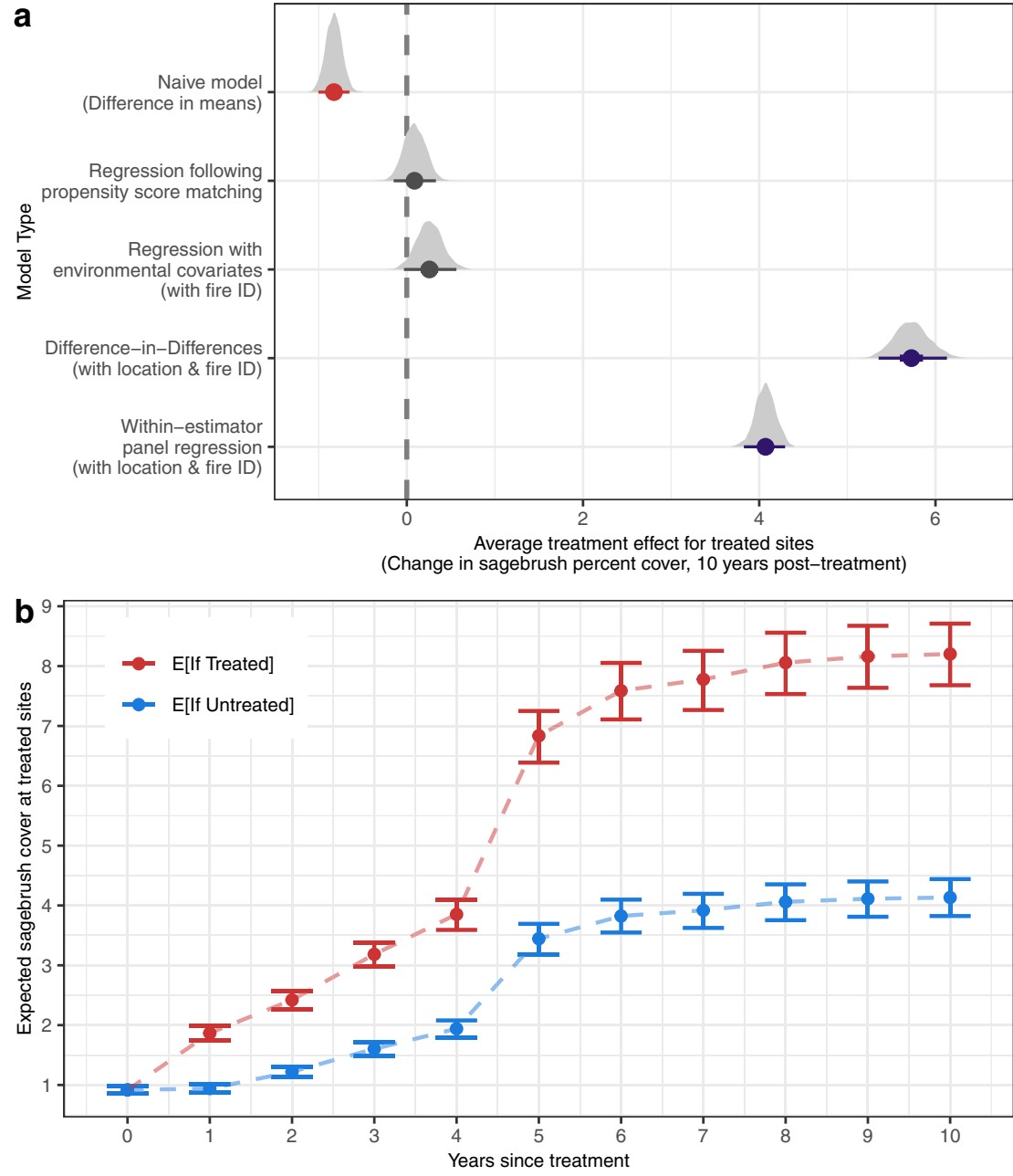

**Fig. 4 Variation in estimated treatment effects among statistical approaches. a** The predicted treatment effects (for treated sites) of post-fire sagebrush seeding on sagebrush cover, identified by five statistical approaches: (1) a naïve model examining only the difference in means between treated and untreated sites; regression analyses that incorporate either (2) propensity score matching or (3) environmental covariates, which considered measured sources of bias; (4) DiD estimation, which considered time-invariant unmeasured sources of bias; and (5) within-estimator panel regression, which additionally considered time-varying unmeasured factors. Dots represent median estimates, lines represent the 95% credible intervals (CIs), and grey density plots indicate full posterior. Red, grey, and purple intervals indicate negative, neutral, and positive estimated effects of restoration treatments on sagebrush recovery, respectively. **b** The expected (E[]) trajectories of sagebrush recovery if treated sites were to receive or not receive seeding treatments, over the 10 years following treatment. Dots indicate median estimates with 95% credible interval bars. Prediction intervals are constructed from posterior draws of the linear predictor for the within-estimator panel model, while holding weather covariates at their means.

ecosystem. Though several frameworks outline recommendations for prioritization of restoration sites in sagebrush steppe[5,45–47], actual treatment implementation may diverge from these conceptual guidelines. Treatment deployment is determined by a complex set of interactions between national-scale (i.e., Department of Interior) allocation of funding, regional-level (i.e., BLM districts) coordination of emergency stabilization and restoration resources, and local implementation (i.e., BLM field offices),

influenced by individual practitioners' interpretations of prioritization guidelines. Further, wildfires occur unpredictably in space and time. Thus, the implementation of restoration may be further shaped by other stochastic or opportunistic factors, including interannual variation in seed availability, contractor availability, timing of funding availability relative to sagebrush phenology, suitability of weather conditions, and social factors that vary across the range of sagebrush steppe. Thus, it is critical

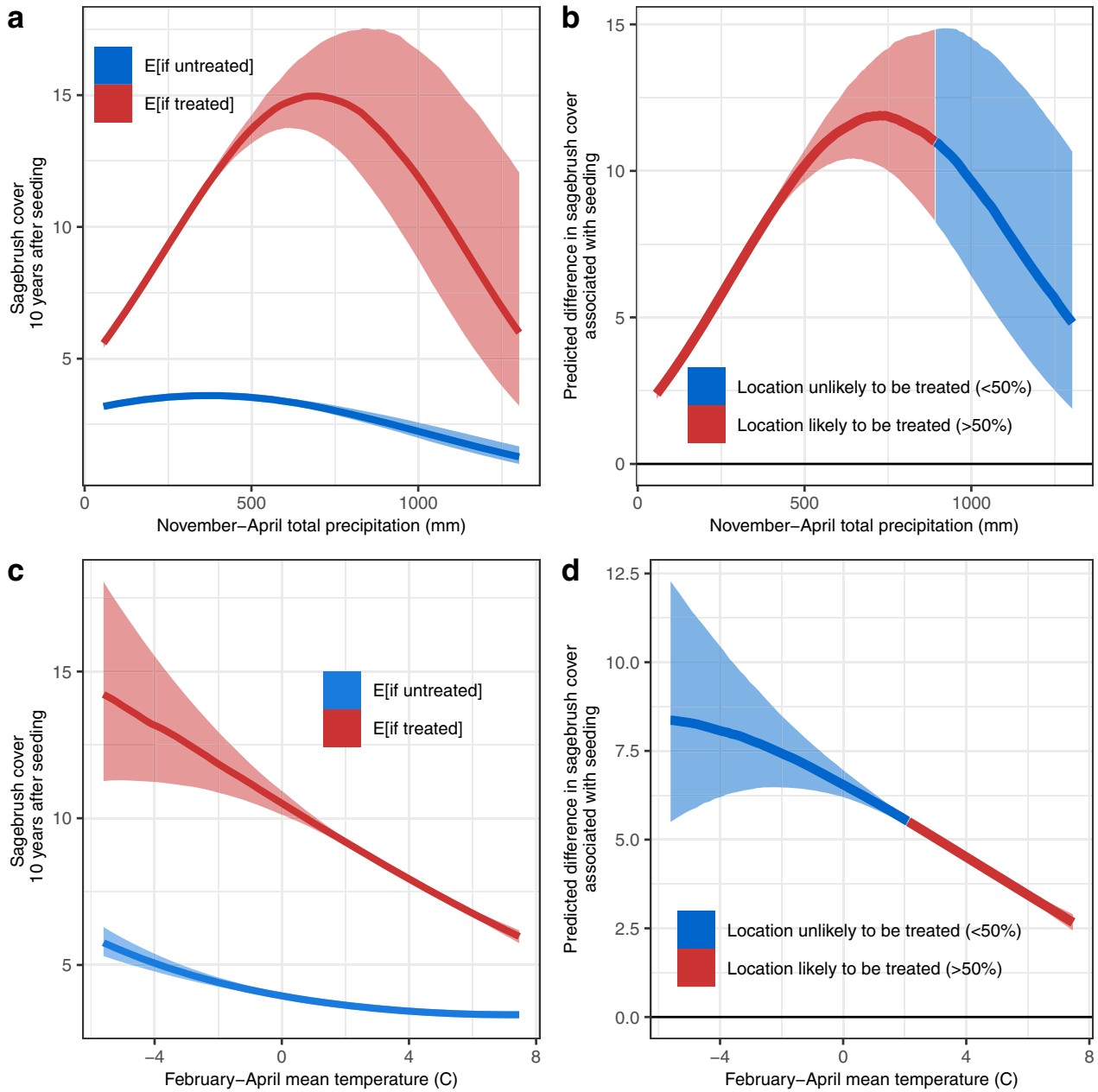

**Fig. 5 Variation in the effects of seeding treatments across climatic gradients.** Panels show sagebrush percent cover (%) 10 years following fire (shown for the Central Basin and Range Ecoregion), along gradients of (**a**, **b**) November-April total precipitation (30-year average) and (**c**, **d**) February-April mean temperature (30-year average). Panels on the left (**a**, **c**) illustrate the expected ("E") of sagebrush cover for sites in the treated group, if they were to either receive or not receive treatment (median expectations, shown with 50% credible intervals). Panels on the right (**b**, **d**) illustrate the difference in sagebrush recovery observed when treatment is applied (median expectations, shown with 50% credible intervals). Positive values (above the solid line at 0) indicate gains associated with reseeding; line color indicates whether restoration is likely (>50% probability of occurrence) or unlikely (<50%) to occur at given precipitation (**b**) or temperature (**d**) conditions, based on the model of treatment probability (see Figs. 2, 3). Posterior parameter estimates for all covariates included in the associated model (n=20,000 locations) can be found in Supplementary Fig. 5.

to document possible gaps between where restoration is recommended and where treatments are actually completed, and then identify whether the locations receiving management represent an efficient use of limited financial, sociopolitical, and biological resources.

**Nonrandom application of restoration treatments.** The shift in treatment effects across our analyses may be best explained by the biophysical and anthropogenic factors that were associated with the probability of a burned location receiving restoration, as identified

in the propensity score model. Over several decades, treatments on BLM land recorded in the LTDL database have been systematically applied in areas characterized by more stressful climatic conditions and smaller pre- and post-fire surviving sagebrush populations, compared to the broader range of conditions in sagebrush steppe ecosystems (Figs. 2, 3). Selection for relatively degraded, dry, and warm sites may also correlate with greater occurrence of fire and more severe annual grass invasion, historically[5].

We hypothesize that these correlates of restoration occurrence may reflect a combination of the perceived environmental

requirements for sagebrush recovery, the narrower range of conditions that may exist on BLM lands, and the cost-benefit relationships faced by practitioners. For instance, restoration practitioners may be less likely to apply treatments in the wettest or coolest sites[35,51] or in areas with the largest pre-fire or surviving post-fire populations (which may exhibit more stable population dynamics[4]), where sagebrush may be perceived more capable of recovering without interventions. Simultaneously, practitioners may be marginally less likely to apply treatments in areas with smaller pre-fire populations, if these areas are perceived to be suboptimal sagebrush habitat. Currently, similar temperature and moisture regime characteristics estimated by the US National Resource Conservation Service soil survey provide guidance for prioritization of restoration treatments[5,45,46]. These regimes have been demonstrated as indicators of post-fire resilience in sagebrush steppe ecosystems, with warmer, more xeric soil regimes resulting in decreased plant community resilience to disturbance and decreased resistance to post-fire cheatgrass invasion[5].

Selection against sites with large pre-fire sagebrush populations may also relate to practitioner perceptions of wildlife attributes or other ecological values in these stands. Critical habitat for listed or candidate threatened and endangered species is a key selection criteria outlined by the BLM Emergency Stabilization and Rehabilitation program, which plans and implements the post-fire seedings we evaluated[47]. Greater sage-grouse (*Centrocercus urophasianus*), a sagebrush-dependent species that has experienced large declines or local eradication across the western U.S. due to loss of the sagebrush-steppe habitat, may preferentially occupy areas of moderate sagebrush cover[55]. Extremely dense sagebrush stands can result from overgrazing and depletion of perennial grasses; when these dense stands with depauperate understories burn, lack of native resprouting perennials can result in increased invasion by exotic annual grasses[3,5,56]. Thus, selection toward sagebrush stands with moderate density before fire may also reflect practitioners' prioritization of seeding investments into known wildlife habitat. Additional surveys, which specifically investigate the preferences, resources, institutional constraints, and ecological criteria that shape decisions made by restoration practitioners and organizations, are required to confirm whether these mechanisms drive biophysical differences in treated and untreated areas.

**Impacts of selection bias on inference of treatment effects.** The shift in treatment effect across statistical approaches examined in this analysis may explain why past evidence for treatment efficacy has been equivocal, especially in large-scale observational studies of seeded post-fire sagebrush habitat[3,33,34,36–39]. The disproportionate allocation of treatments toward sites that are more climatically stressful, more degraded by fire, or generally more ecologically "difficult" to restore (Figs. 2, 3) generated a negative correlation between restoration and post-treatment sagebrush cover in the "naïve" null model (Fig. 4), which did not account for differential biophysical characteristics of treated and untreated sites.

Yet, analyses that considered observed environmental sources of selection bias (via propensity matching or multiple regression) also did not detect clearly positive treatment effects, even after the inclusion of a set of biophysical characteristics that have been frequently integrated into past observational studies of restoration efficacy in sagebrush steppe ecosystems[34–36,44,51,52]. Regression including environmental covariates or following propensity score matching most closely reflect commonly applied approaches in large-scale assessments of sagebrush recovery. However, positive effects of restoration were detected only after applying DiD and

within-estimator panel regression methods, which considered either time-invariant or time-varying unobserved sources of heterogeneity. This suggests that quasi-experimental designs that match sites based on similarity in observed characteristics (e.g. refs. [34,36]) may underestimate effects of restoration if additional, unobserved characteristics differ between treated and untreated sites.

Ecologically relevant but unobserved sources of bias in our analyses may include finer scale environmental factors, such as soil microsite conditions, or biotic interactions influencing treatment allocation. Mean temperatures, precipitation, soil texture and water retention (i.e. soil percent clay), and late-winter snowpack retention have all been correlated with survival and growth of sagebrush in past field studies, suggesting that soil-water availability in the early stages of sagebrush development, when seedlings are most susceptible to drought, may play a key role in restoration efficacy[35,51,52,57]. While the spring temperature, spring precipitation, and soil variables that we included in our analyses may influence the phenology of spring snowmelt, soil moisture, and recharge, they may not correlate perfectly with fine-scale soil-water potential. Another ecologically relevant, but omitted factor was systematic differences in cheatgrass invasion and biotic interactions at treated and untreated sites; for instance, competition with invasive cheatgrass determine trajectories of sagebrush recovery and further alter soil water availability[58]. Similarly, pre- and post-fire livestock grazing, which varies in pressure across BLM lands, may correlate with treatment application and sagebrush recovery[59]. However, to our knowledge, no existing datasets accurately summarize grazing history across this broad spatiotemporal extent.

DiD estimation, which considered both observed and unobserved time-invariant sources of bias associated with systematic differences between treated and untreated groups, suggested that post-fire seeding treatments generated small (5.7%) improvements in sagebrush cover in treated areas (Fig. 4). The estimated effect of reseeding on sagebrush recovery in treated sites slightly decreased using within-estimator panel regression approaches (4.1%), likely due to the consideration of time-varying covariates, such as spring weather conditions. Weather (i.e. interannual variation in climate) and other "year effects" are commonly discussed potential drivers of variable outcomes in restoration[2,60–64], including in sagebrush systems[52,65,66]. Time-invariant mean climatic variables imperfectly capture the specific weather conditions that occurred in the years following restoration. Further, the extent of interannual climatic variation may differ between hotter, drier and cooler, moister sites, suggesting that the strength of weather fluctuations may also correlate with treatment assignment.

We propose that the within-estimator panel regression estimate, which incorporates time-varying and time-invariant sources of heterogeneity, represents the least biased estimate of the effect of post-fire seeding in sagebrush steppe ecosystems. However, the validity of each approach and associated estimated treatment effect rests on important assumptions and limitations. Regression following propensity score matching assumes that, after matching based on measured covariates, treatment assignment will be effectively randomized; given the comparative shift in treatment effect in matched, DiD, and within-estimator panel regression approaches, this assumption is likely poorly founded, based on the set of biophysical covariates incorporated here[34–36,44,51,52]. In contrast, DiD and within-estimator panel regression methods are limited by the assumptions that in the absence of treatment, the differences between treated and untreated groups are constant, that the predictors are exogenous, and that treatment effects are static over time[26,67,68]. In particular, we expect treatment effects to be heterogenous across

the range of sagebrush steppe and explored this variation in a subsequent form of our DiD model (Fig. 5).

**Variation in restoration efficacy across heterogeneous landscapes.** Recent discussions of restoration efficacy in sagebrush steppe ecosystems have called for flexible management, such as planning that can integrate near-term weather forecasts to capitalize on years with particularly favorable conditions for establishment[65]. Several past studies have identified factors associated with the probability of sagebrush establishment in burned areas that were seeded, finding that sagebrush occupancy was more likely at higher elevation and in areas with moister, cooler spring soil conditions[35,51]. However, these analyses have typically focused on variation in sagebrush recovery exclusively within treated areas, rather than the extent to which gains achieved by restoration (compared to untreated areas) vary across these biophysical characteristics of sites (though see refs. [6,34,38]).

DiD estimation suggested that post-fire seeding treatments increased sagebrush cover by an average 5.7% in treated areas across this study's extent. However, in a DiD model that incorporated interactions between treatment occurrence and site characteristics, the gains in sagebrush cover achieved by restoration varied substantially across climatic gradients (for instance, ranging from 2–12% increases in sagebrush cover for the Central Basin and Range ecoregion), as illustrated by the interaction terms between the DiD estimates and biophysical variables in subsequent analysis (Fig. 5, Supplementary Fig. 7). Similarly, a field study of seeding efficacy following prescribed burning in Oregon western juniper-big sagebrush stands within the xeric-frigid temperature-moisture regime found that climate (as affected by aspect) significantly altered the strength of treatment effects[38]. We propose that flexible management plans may use similar analysis of observational datasets to predict where treatment effects will be greatest (in contrast to where plant population recovery is most likely) to expend limited restoration resources most efficiently, given intensifying species invasions, rapidly changing climates, and accelerating fire cycles[5,30,49,50]. Over the historical range examined in this study, restoration was applied in many areas that were predicted to experience large gains in sagebrush cover; however, treatments were also likely to occur in areas where treatment effect sizes were minimal (Fig. 5b, d).

This variation in treatment effect sizes along gradients of precipitation and temperature (Fig. 5, Supplementary Fig. 7) may further explain why past evidence for the efficacy of post-fire seeding efforts has been mixed. The extent to which past studies have detected strong, weak, or neutral improvements in sagebrush cover associated with restoration can depend upon the range of climate characteristics observed in a particular dataset. Further, the overall pattern of selection bias observed in this study–in which restoration was more likely to occur at more stressful sites (Figs. 2, 3), thereby reducing the perceived effects of restoration (Fig. 4) -- may be common across a wide variety of focal systems in restoration ecology (e.g. deforested areas of the tropics[14]), given limitations associated with land ownership or the obvious need for practitioners to make strategic decisions about the allocation of limited management resources.

**Quantifying restoration treatment effects.** Massive restoration efforts, such as post-fire seeding of dryland shrubs, span regions and encompass environmental variability at a scale that is difficult to capture in traditional smaller control–impact studies. Quasi-experimental approaches are not intended to be a replacement for planned, randomized field experiments. However, long-term observational datasets, including those derived from satellite

imagery, are increasingly accessible to researchers and may represent powerful tools for both improving management and advancing restoration theory, while capturing broad spatio-temporal extents[2,12]. As we have demonstrated, if sources of bias (including those introduced by humans' perceptions of or preferences for restoration sites) are not statistically incorporated when working with non-experimental datasets, correlations between treatment application and the characteristics of sites may cloud inference of restoration efficacy under both changing climatic conditions and anthropogenic pressures.

Each of the inferential approaches in this analysis has comparative limitations, and we caution against the application of any single treatment effect or singular adoption of any statistical model. However, this analysis of remotely sensed data clearly illustrates that selection bias may be a key hurdle to understanding and predicting treatment effects across a broad spatial and temporal extent and suggests that post-fire seeding may be more broadly beneficial than previously documented. Quasi-experimental approaches are regularly applied in studies examining the ecological or socioeconomic impacts of other public environmental programs (such as wildland firefighting efforts[69], information campaigns to reduce water consumption[70], and payments or land protections to prevent deforestation[18]); however, they have been noticeably absent from large observational studies of restoration outcomes[14], suggesting a need for interdisciplinary research that examines social and ecological motivations that underlie the nonrandom application of restoration treatments.

Analytical approaches that consider selection biases may be essential tools for both adaptive management and identifying drivers of restoration efficacy across these large landscapes and timeframes. We propose that the results summarized here illustrate key considerations for examining the breadth of environmental heterogeneity that face landscape-scale land management experiments in restoration ecology.

## Methods

We compared the estimated efficacy of sagebrush reseeding efforts identified by four statistical approaches: (1) a naïve "null model" comparing the mean sagebrush cover at treated and untreated sites, in which neither observed or unobserved sources of selection bias are considered; (2) analyses that incorporated observed sources of selection bias, using one of two approaches (propensity score matching and a regression containing environmental "control" covariates); and (3) using Difference-in-Differences (DiD) estimation to consider time-invariant sources of unobserved and observed bias; and (4) using within-estimator panel regression to consider unobserved time-invariant and timepoint-specific heterogeneities, along with measured time-varying covariates. We then identified apparent environmental drivers of where restoration treatments are implemented and compared the gains in estimates of sagebrush cover achieved by post-fire seeding efforts along gradients of several biophysical characteristics of sites that shape selection of post-fire treatment locations.

**Study system.** The sagebrush steppe ecosystem is defined by presence of one or multiple taxa within the big sagebrush complex (*Artemisia tridentata*), with lesser occurrences of three-tip sagebrush (*Artemisia tripartita*), low sagebrush (*Artemisia arbuscula*), or other *Artemisia* species. Herbs, perennial bunchgrasses, or annual grasses can dominate in these ecosystems as a result of historic fire, some types of grazing, or management actions such as herbicides and seeding[56]. Fire severity can be highly variable, ranging from complete removal of aboveground biomass over large areas, to mosaics of unburned patches within more severely burned areas. Post-fire restoration seedings of sagebrush typically occur either in the fall or early spring following a wildfire via ground broadcast seeding or aerial broadcast from aircraft. Sagebrush seeds germinate in early spring, during the transition from low temperature to water limitation[71]. Many studies have sought to understand the environmental drivers of sagebrush's post-fire establishment and population recovery, primarily focusing on climate and weather effects during this critical spring period. In this arid ecosystem, site elevation, temperature and precipitation[10,52,72], spring soil water potential (as mediated by antecedent snow-pack, spring temperatures, and spring precipitation;[35,51,57]), and the occurrence of spring freezing events[73] have all been associated with sagebrush establishment, survival, or growth.

**Data sources**. We compared post-fire sagebrush recovery for treated and untreated locations across the Great Basin, identified using historical fire perimeters, the National Land Cover Databases' historical sagebrush cover product, and the Land Treatment Digital Library, a catalog of management actions on BLM lands across the western U.S. (LTDL;[48]). Using the LTDL, we extracted 10,000 randomized locations, in which at least one (and up to four) *Artemisia* species or subspecies was seeded (via aerial or ground methods) in the fall or spring following the fire. These burned, treated areas were identified by the overlap between treated and fire perimeter polygons from 1985 to 2005 (Monitoring Trends in Burn Severity;[74]). Areas that burned more than once during the focal time period were excluded from the analysis, and burned locations were randomly selected using the spsample() function in the *sp* package in R[75]. We then randomly extracted a series of 10,000 burned, untreated locations using the same fire perimeter polygons. The randomly selected set of treated and untreated datapoints were located within the perimeters of 1539 fires occurring between 1986 and 2001.

Sagebrush cover before and after fire was extracted at 30-m pixel resolution from the Rangeland Condition Monitoring Assessment and Projection dataset (RCMAP, formerly known as the National Land Cover Database's Back-in-time Sagebrush Rangeland Fractional Component), which is based on Landsat, Quickbird, and AWiFS imagery for each year from 1984-2018 across the Great Basin[8,11]. Using these spatial locations and temporal information about the year in which treatments occurred, we extracted the estimated sagebrush cover for each location for: (1) the year preceding wildfire to establish the site's pre-fire sagebrush population size; (2) one year post-fire to establish the "pre-seeding-treatment" population size (i.e. sagebrush cover surviving wildfire, as sagebrush do not recovery rapidly enough to generate a signal to satellites within the first post-fire year); and (3) each of the 10 years following treatment to quantify the "success" of reseeding treatments. We selected a 10 year timepoint to ensure that the stand was beyond the point at which the remotely sensed signal for cover was likely to fluctuate strongly (previously identified as ~6 years after fire)[76]. The results of our analysis did not qualitatively differ if we examined longer (15-year) or shorter (8-year) timepoints (Supplementary Figs. 13, 14). Based on field and satellite-based validation studies, the shrub and sagebrush components of RCMAP have an out-of-sample $R^2$ of ~0.60[77]. To ensure the analysis was specific to sagebrush habitat, all extracted locations were within the spatial extent of the RCMAP sagebrush cover product and contained at least 1% estimated sagebrush cover in at least one year during the five years preceding the recorded wildfire. See Supplementary Note 5 for additional information about RCMAP validation, performance, and limitations.

We identified biophysical characteristics that we thought may influence treatment location or restoration success, based on covariates that appear in recent frameworks for the prioritization of sagebrush restoration sites or variables that have been commonly included as "control" variables in previously-published studies of post-fire recovery of sagebrush[34,35,51,52]. For each location, using the *raster* package in R[78], we extracted or calculated elevation[79], soil percent clay and sand (using a product aggregating the USDA-NCSS SSURGO and STATSGO datasets)[80], heatload[81], the U.S. Environmental Protection Agency's Level III ecoregion, fire size, and distance to a major road from the U.S. Census Bureau's TIGER database. We calculated November-January (winter) and February-April (spring) mean temperatures and mean total precipitation for the period for between 1984 and 2014 for each location, using the gridMet modeled meteorological dataset[82]. GridMet contains daily, high-spatial resolution (4-km) climate estimates for the contiguous U.S. from 1979 to present.

**Propensity score matching**. Propensity score matching (PSM) attempts to estimate an accurate treatment effect by accounting for included measured covariates that may influence the probability of an observation receiving the treatment, as these covariates can generate bias if they also influence the response variable. Propensity score matching uses logistic regression to estimate each observation's probability of receiving treatment based on the suite of observed covariates, and then pairs treated and untreated observations with similar treatment probabilities, including only matched sets below a given threshold for similarity, defined by the investigator. This approach relies upon the assumption that matching simulates randomization of treatment assignment[20,83,84]. Further, because PSM relies on narrowing observations to a region of common support between groups, the treatment effect estimated from this model may focus upon a narrower range of possible conditions, relative to the "unmatched" dataset.

We matched treated and untreated observations based on their propensity for receiving treatment, using the *MatchIt* R package's nearest neighbor method[85] (See Supplementary Note 1 for additional information about the matching algorithm). In the propensity score model, we included the following biophysical variables: sagebrush cover 1-year pre-fire, surviving (post-fire) sagebrush cover, total November-April precipitation, February-April mean temperature, fire size, soil percent clay, soil percent sand, elevation, distance from road, and ecoregion identity. We selected these variables to reflect, as closely as possible, the biophysical factors that are commonly included in past frameworks for restoration prioritization or studies of post-fire sagebrush recovery[34–36,44,51,52], as means of controlling for systematic differences between treated and untreated sites or variation within treated areas. Quadratic effects were included for the temperature, precipitation, elevation, heatload, distance from road, and pre-fire sagebrush cover

variables as we hypothesized that intermediate levels of these variables may be preferentially selected by managers. For instance, restoration treatments may be disproportionately applied to sites with intermediate precipitation levels if restoration efforts are likely to be unsuccessful at low precipitation sites and if high precipitation sites are perceived to be more capable of recovering without intervention[5]. To ensure that the matching protocol would sufficiently limit observed sources of bias, we limited the distance between pairs' propensity scores, using a "caliper width" criteria (which determines the maximum allowable difference between paired sites) equivalent to a fifth of the standard deviation of the mean propensity score[86]. The matching protocol resulted in 11,012 paired treated and untreated pixels in the reduced dataset, within 940 fires.

To infer relationships between sites' biophysical characteristics and treatment probabilities, we also constructed a Bayesian Bernoulli glmm (with a logit link) containing treatment occurrence as the response variable and each of the covariates included in the matching procedure as independent variables, with a varying intercept for ecoregion. The full list of variables included in this model is displayed in Fig. 2b.

**Comparison of treatment effects estimated by differing statistical approaches**. To compare how inferences on restoration efficacy may vary depending on how selection biases are considered, we developed five Bayesian negative binomial generalized linear models and mixed models with sagebrush cover 10 years following fire as the response variable (summarized in Table 1). We adopted negative binomial errors because the sagebrush cover from RCMAP are positively constrained integers. Although our response variable is also bounded at 100, graphical posterior-predictive checks (Supplementary Fig. 4) indicated that the predictions generated from the model reproduced the overall pattern in the raw data and did not result in predictions exceeding 100.

The first analysis modeled the full, "unmatched" dataset as a function of a single fixed independent variable for whether an individual location received treatment (see Table 1). Though simplistic, this "naive" model was designed to quantify the difference in means between treated and untreated groups without accounting for sources of bias, as a conceptual null model.

The second and third models accounted for measured sources of bias, either analyzing the subset of locations remaining following the propensity score (referred to as the "matched" dataset) or analyzing the full, unmatched dataset, using a regression that contained environmental variables that represented factors that have been commonly incorporated as "control" covariates in past studies of sagebrush recovery[34–36,44,51,52]. These environmental factors included heatload, elevation, soil percent clay, soil percent sand, November-April total precipitation, February-April mean temperature, ecoregion, and fire identity (structured as a varying intercept, to account for spatial and temporal variation between sites)[34–36,44,51,52].

A key limitation of using "control" variables or propensity score matching is that it only addresses observed sources of bias in treatment location, which the researcher is both aware of and has measured covariates for. Difference-in-differences and within-estimator panel regression approaches may overcome this limitation by comparing observations from before and after a treatment was applied to remove omitted sources of time-invariant and time-varying bias, respectively. DiD uses a binary variable for whether a treatment was applied to a given individual or location to account for persistent differences between treated and untreated groups (group), and a binary variable for whether an observation describes the status of an individual or location before or after treatment (time). The effect of the treatment is identified using the parameter estimate associated with the interaction between group and time variables. Within-estimator panel regression extends this structure to incorporate multiple post-treatment observations, with time-varying indicator variables that capture unobserved heterogeneity that is distinct to each observation timepoint.

To assess how accounting for unobserved sources of bias may influence inference of restoration success, we developed multilevel DiD and within-estimator panel regression models, which examined sagebrush cover (Models 4 and 5 in Table 1). For the DiD model, the response was a single observation of sagebrush cover 10 years following treated, while the within-estimator panel regression modeled annual cover observations across this same 10 year period. Each model contained: (1) an indicator variable for whether the observation was in treated or untreated groups; (2) whether the treatment had occurred by the observed timepoint; (3) and a variable for timepoint. In the DiD model, time was a binary indicator, while the panel regression model included "time since treatment" (with year 0 representing the observation before treatment occurred, and post-treatment observations from year 1 to 10) to represent unobserved heterogeneity specific to each year's observed sagebrush cover. Structuring time as an indicator for time-since-treatment allowed us to avoid comparing recently-treated sites with trajectories in previously-treated sites, given that seedings were deployed across the two decades in our study[67]. Both models contained a varying intercept for location identity (to account for unmeasured differences between sites and autocorrelation between repeated measures) and a varying intercept for the identity of the fire impacting the site (to account for possible impacts of spatiotemporal autocorrelation on estimation of the treatment effect). The panel regression also contained a time-varying covariate for each observation year's spring mean temperature and total precipitation to assess the extent to which these annual

deviations from time-invariant mean climatic conditions further impacted sagebrush recovery.

In nonlinear DiD and panel models, the indicator variable for treatment occurrence in a particular timepoint identifies the "incremental" treatment effect, which shares a sign with the treatment effect, but which must be interpreted differently than the interaction term in a linear DiD model[87]. Thus, we calculated the treatment effect as:

$$= \exp(group + time + group * time) - \exp(group + time) \quad (6)$$

A key assumption of DiD regression is that the treated and untreated groups exhibit parallel trends before the treatment is applied. To confirm the assumption of parallel trends, we visually compared the population trajectories of treated and untreated locations for the 10 years preceding wildfire (Supplementary Fig. 1). In addition to incorporating a varying intercept for fire identity to account for spatial and temporal aggregation of observations in the dataset, we diagnosed models for evidence of additional spatial autocorrelation in their residuals using a Moran's I correlogram and visual inspection of a spatial variogram (Supplementary Figs. 8–10).

**Variation in treatment effect size across biophysical gradients**. We developed an additional DiD model to assess how treatment efficacy varied across biophysical conditions that may be of interest to restoration practitioners across the Great Basin. The model contained interaction terms between the DiD interaction term (time*group) and three biophysical characteristics of sites, including February-April mean temperature, November-April total precipitation, and soil percent clay, to identify how gains in sagebrush cover achieved by post-fire seeding varied under different conditions. Though other variables may influence treatment success, we selected these covariates based on their relevance in the propensity score analysis and to the climate mechanisms highlighted in past studies of sagebrush recovery[10,35,51,52,72,88].

In all models, continuous variables were centered and scaled by 1 standard deviation. Models were estimated using the language Stan and the *brms, rstan*, and *cmdstanr* packages in R (Version 4.1.1), using weakly informative priors and a Markov Chain Monte Carlo sampler with four chains, each with 2000 iterations and 1000 warmup iterations[89,90]. We assessed effective sample size and model convergence, indicated by Gelman-Rubin statistics close to 1 and stable-well mixed chains[89]. Parameter estimates with 95% credible intervals that did not contain zero were considered to have "non-zero" effects on the response variable[89]. Model fit was assessed by calculating the mean absolute error and using graphical posterior predictive checks, which compare the observed data to draws from the posterior predictive distribution generated by the model (Supplementary Fig. 4). Posterior predictions from models and visualizations of model uncertainty were generated using the tidybayes and bayesplot packages[91–94].

**Reporting summary**. Further information on research design is available in the Nature Research Reporting Summary linked to this article.

## Data availability

This analysis was developed using the Land Treatment Digital Library (https://ltdl.wr.usgs.gov/), the Monitoring Trends in Burn Severity historical fire perimeters (https://www.mtbs.gov/), GridMet surface meteorological data (https://www.climatologylab.org/gridmet.html), the U.S. Environmental Protection Agency's Level III ecoregion categorizations (https://www.epa.gov/eco-research/level-iii-and-iv-ecoregions-continental-united-states), and the University of California Davis' Soil Resource Lab's Soil Properties application (https://casoilresource.lawr.ucdavis.edu/soil-properties/). The processed set of treated and untreated pixels (from which our locations were randomly selected) and source data for figures are provided via Dryad: https://doi.org/10.25338/B8W63R[93].

## Code availability

All code used to develop the analyses presented in this manuscript can be found at https://doi.org/10.5281/zenodo.6565074[94].

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

## Acknowledgements

We thank Cara Applestein for management of the RCMAP data products, and members of the USGS Forest and Rangeland Ecosystem Science Center for initial organization of sagebrush seeding treatments from the Land Treatment Digital Library. Collin Homer and Matt Rigge provided access to and interpretation of the RCMAP data. We thank Robert Shriver for feedback that greatly improved this manuscript. Funding was provided by grants to M.J.G. from the Southwest Climate Science Adaptation Center (CASC) with additional contribution from the Northwest and North Central CASC on Ecological Drought, in addition to the Joint Fire Science Program. A.B.S.W. was partially supported by the National Science Foundation Postdoctoral Research Fellowship in Biology (award number 2010868, A.B.S.W.) and the NSF Idaho EPSCoR Program (award number OIA-1757324). Any use of trade, product, or firm names is for descriptive purposes only and does not imply endorsement by the U.S. Government.

## Author contributions

M.J.G. and A.B.S.W. co-developed the research questions and the subsequent manuscript. A.B.S.W. conducted the analysis with input from M.J.G.

## Competing interests

The authors declare no competing interests.
