## [Peer Review File · Nature Communications]

Statistical considerations of nonrandom treatment applications reveal region-wide benefits of a widespread post-fire restoration actionREVIEWER COMMENTS

Reviewer #1 (Remarks to the Author):

This paper describes a modeling effort to predict the effect of sagebrush seeding on post-wildfire sagebrush recovery in the western United States. The authors use propensity matching to show that seeded areas have a different bioclimatic/geographic profile than unseeded areas, demonstrating that restoration practitioners deployed seeding with some bias, presumably for sites where they thought that seeding was more likely to assist sagebrush recovery. A model that ignored this bias suggests that seeding had a negative impact on sagebrush recovery, which is counterintuitive. Using only a subset of samples with similar bioclimatic/geographic profiles, the same model suggested that restoration had no effect on sagebrush recovery. A different type of model (DiD) that accounted for change in sagebrush cover pre-treatment to post-treatment suggested that seeding had a small positive effect on sagebrush recovery. The authors use this final model (actually one of two DiD models with and without a random intercept for fire identity) to demonstrate where seeding had the greatest impact on increasing sagebrush cover.

This paper is one of very few that addresses the issue of selection bias in ecological restoration head on. Moreover, it does so at a spatial scale relevant to the large-scale restoration that is needed to move the needle on biodiversity conservation, climate change mitigation, preventing desertification, and provisioning other ecosystem services. The modeling approach seems like it could (and should) be used in a variety of other ecosystem types to evaluate the true impact of large-scale restoration activities. Such evaluations will be increasingly important over the next decade as restoration funders seek information on the efficacy of their investments, particularly for carbon storage. As such, this well-written paper merits a high-visibility platform.

Major Comments

1. This study's biggest weakness is that the magnitude of difference in sagebrush cover between seeded and matched unseeded sites was small, +1.1% on average (ranging up to +8.1% in some contexts). This relatively small difference makes it hard to draw a strong conclusion about the efficacy of seeding, particularly when the average difference is exceeded by the measurement error in the RCMAP dataset (6%). Is +1.1% meaningful? Is it worth the expenditure (>\$100M USD since 1990)? The authors are wise to word their conclusions conservatively, without making a clear policy recommendation about sagebrush seeding in the western US, other than to say that the net effect is positive. I do not have a recommendation for addressing this comment other than perhaps to focus more of the discussion on the implications of this work beyond the study system. In my view, the greater importance of this paper is in its novel methodological approach rather than the implications for U.S. restoration policy regarding sagebrush.
2. A second weakness, which the authors describe in the discussion, is that some potentially important predictors were not included in the propensity matching and subsequent modeling. The most striking omissions are substrate (e.g., soil type), land use (e.g., grazed versus not), and within-year environmental conditions (e.g., spring soil moisture in the year of seeding). Clearly not every possible variable can be included in this modeling effort, and some variables may simply not be available, but what is the rationale for excluding these particular variables? I think that the effect of within-year environmental conditions on sagebrush seeding success would be especially interesting to restoration practitioners in the sagebrush system.

Minor Comments

3. Line 124. I would have benefited from a more thorough description early on of DiD modeling, which I am unfamiliar with.
4. Line 289. If I understand correctly from the methods, DiD modeling is a clever way of specifying and interpreting a GLM or GLMER to assess change in the response variable (via the treatment \times time interaction) while still ultimately predicting the response variable's state. Wouldn't it be simpler and accomplish the same thing to model sagebrush cover's change over time in treated vs. untreated areas instead? If not, why?
5. Paragraph starting at Line 379. Could you provide a little bit more information about the RCMAP? How is this product created?

6. Line 388. For someone outside of the sagebrush system, it's not clear how 10 years is a relevant time period for sagebrush life history. Is this how long sagebrush seeds remain viable in the soil seedbank? A citation would be helpful.
7. Line 393. Why is "sagebrush cover forced to be the same or greater than the previous year in burned areas?" I don't understand why this is necessary. Can't sagebrush cover decline over time too?
8. Line 463 (equation). "Treatment" and "Group" seem to be used interchangeably between this equation and the preceding paragraph. It would be better to use one throughout.
9. Figure 5. What are the units on the y axis? The trend is clear, but it's not clear what the magnitude or importance of the pattern is.
10. Figure S1. The legend says "post-fire" but the caption and the x-axis label both imply that these trends are supposed to be pre-fire.

-Leighton Reid

Reviewer #2 (Remarks to the Author):

Selection bias in restoration application is an issue when observation studies attempt to draw conclusion from restoration efforts. The idea that selection bias exists in restoration practices is not novel as restoration is not applied at random. The implications of this study are very important and identify some pitfalls with using observations studies to evaluate restoration and other natural resource management practices. The manuscript could be strengthened by including relevant literature on successful sagebrush seedings that contrast the observation studies results to provide a more complete framework for the study and place the results of this study in context with the literature. The study is well-design, sound, and of importance to the field of restoration and land management. The conclusions and interpretations are supported by the data.

Ln 22-23 this is not unexpected. Restoration would be more likely needed in more stressful and degraded sites. This could be worded into a hypothesis of the study.

Ln 85 tree invasion has also been a major contributor to the decline of *Artemisia* species in western North America

Ln 92-106 The authors need to include research projects (Davies and Bates 2017; Davies et al. 2018, 2019; Urza et al. 2019) that have reported successful sagebrush restoration. This would better frame their argument that observation studies of restoration efforts may have selection bias as well as give a more complete picture of sagebrush restoration variability.

Davies et al. 2018. *Rangeland Ecology & Management* 71:19-24.

Davies and Bates. 2017. *Restoration Ecology* 25:33-41.

Davies et al. 2019. *Restoration Ecology* 27:120-127.

Urza et al. 2019. *Biological Invasions* 21:1993-2007.

Ln 110-112 This hypothesis is rather obvious. Undoubtedly restoration treatments are not applied at random. It would defy logic to apply restoration treatments at random. Restoration treatments are applied to areas that will in all likelihood will not recovery on their own. When climate and biophysical traits are more favorable, these sites are allowed to naturally recover. Possibly modify the hypothesis to be more specific or remove it.

Ln 156 Northern what?

Ln 207 The authors need to place their results in context with several recent studies that have found higher success with seeding sagebrush than older studies.

Ln 225-227 cite literature showing that sagebrush natural recovery is greater in the cooler and

wetter sites

Ln 241 "threatened" is a legal term that I don't think applies to greater sage-grouse at this time.

Ln 253-254 Need to place a caveat at this point that this is widely known (i.e. not a surprise to anyone that is familiar with restoration practices in drylands).

Ln 262 what about the studies that have detected an increase in sagebrush cover with restoration?

Ln 268-270 Also aspect and microsites

Ln 274 existing vegetation also influences soil water. Cheatgrass depletes soil water earlier than native vegetation

Ln 293 could minor losses from restoration be attributed to seeding of other species (perennial grasses)?

Ln 353 use citation number compared to name and year

Ln 354-355 combustion of herb understories and survival of shrub overstories rarely, possibly almost never, occurs in sagebrush communities. Herbs act as ladder fuel to engage highly combustible sagebrush

Ln 355 and 358 include citations

Ln 558 citation format is different than other citations

Ln 599 citation is lacking journal, volume, and page number

Reviewer #3 (Remarks to the Author):

Simler-Williamson & Germino revisit the benefits of post-fire re-seeding efforts on sagebrush recovery with a particular focus on the potential for selection bias to confound statistical estimation. They compare a naïve statistical model with models that address selection on observables (propensity-score matching) and selection on unobservables (difference in difference). It is an interesting application of causal inference methods. However, I have several important methodological concerns related to all of the models the authors present.

1) With regard to the naïve statistical model, no details are provided regarding the model specification. For example, what biophysical covariates were included, did they account for the year of the fire, the ecoregion, how many different species were seeded? Or was this a t-test ignoring everything except treatment? It would be far more useful if the "naïve" model was based on a prior publication so readers could assess how impactful the PSM and DiD approaches are relative to current understanding in the field.

2) With regard to the propensity score matching, little information is provided regarding balancing diagnostics, how time-varying treatments (ie seeding in different seasons of different years) were considered, or finer details regarding the matching algorithm (was it with replacement or 1:1, if with replacement, how was the presence repeated observations accounted for in the errors). If it is not clear if all of the variables influencing the treatment decision are included, and the argument is not explicitly made that they are (the defining assumption of the method).

3) With regard to the difference-in-difference, the interpretation of the interaction term is not easily translated from a linear model to the nonlinear model used here, and the model does not account for group means in the way it would if the model was linear. This should be addressed.

Additionally, what is the methodological justification for using DiD on the PSM matched subsample? If selection bias only affects the pre-period, it wouldn't be an issue with DiD, it would have to affect the trends. Results from the unmatched sample are not included in the results figures, and it is not specified what standard errors the authors use to account for serial correlation of the errors in the DiD.

Line-by-line comments

Line 70-76: It would be better to phrase these methods with respect to the problem they address and the key assumptions. One is assuming variables driving selection are all observable and measured, which is a high bar.

Line 152-157: Is there not a record or protocol of how seeding locations are chosen?

Line 161: To support the claim that covariates are similar in the matched subsample, please include a summary stats table (or figure) with biophysical characteristics for treatment and control (separately) in the full sample, and in the PSM sample.

Line 167-171: What was the naïve model? Effectiveness of re-seeding is a common research question, please show the results you'd get using a model from the published literature, and then the improvement with PSM and DiD. Presumably the state of the science isn't comparing seeded unseeded without controlling for anything, or if it is, it is comparing sites within the same fire, no?

Line 190-197: Is satellite data across the two decades of this study equally likely to misclassify sagebrush? That might be a tough argument. At least provide evidence if that is the argument. If it does change, it seems like a potential weakness to discuss in the discussion. Also clarify that "repeated measures" only applies to your DiD results.

Line 217: The parenthetical statement is concerning. Please explain when a site is or isn't recorded.

Line 249-250: Do your empirical results reflect knowledge of the treatment decision process from the BLM plans? Do the BLM plans spell out how the treatment decision is made?

Line 253-257: The purpose of this paper is getting confusing. Is it to isolate whether seeding is an effective restoration strategy or is it to understand underlying determinants of seeding by the BLM? If the latter is a part of it, do you learn something from the analysis that isn't outlined in the BLM plans? In other words, it is apparent from the empirical analysis that something besides what the BLM outlines as important determines where seeding occurs?

Line 266-271: Why wasn't soil quality included in the PSM if it is a known and observable (e.g. SSURGO) drivers of treatment and outcome? The underlying assumption of PSM is that all information that can influence the probability of treatment is included (such that conditional on these observable covariates, treatment is as good as randomly assigned).

393: More info needed on RCMAP data accuracy and methods, at least in the SI.

405-407: Clarify wording

407: Rather than jumping into the empirical methods, first explain the major underlying assumption of PSM and its meaning.

414-416: Is this based on the BLM planning document alluded to in the discussion?

417: Provide all variables considered in the SI.

417: Collinearity in the propensity score is of much less importance than satisfying the assumptions of conditional independence. You aren't making statistical inference with the propensity score, rather just interested in the predicted values of the model. 0.5 seems overly strict given the lesser importance of collinearity here. Please try other version of the pscore model.

427: Did covariates balance? Please provide what diagnostics were tested and the results. A summary stats table would be helpful here (the same one as requested in the above comment).

428-434: Please provide more details on the matching algorithm. Was it NN with replacement or without. If repeated, did you adjust for repeated observations? Was it optimal or greedy matching to select the nearest neighbor. This info can be in the SI, but should be available.

437: What is the methodological concern for doing DiD on the matched sample rather than the entire sample? Selection would need to affect the trends across time not just the pre-period to be an issue. It isn't clear to me how that is plausibly occurring, please clarify. Please include the DiD on the unmatched sample in the results figure.

444-447: The interpretation of the DiD interaction term isn't the same in non-linear models as it is in a linear model. The assumptions required for interpreting the interaction term as the DiD in a nonlinear model negate the utility of using the DiD in the first place and don't reflect the logic used here to justify the use of the DiD (lines 450-457). See Lechner 2010 (<http://dx.doi.org/10.1561/08000000014>). This is an important problem to fix for all of the DiD models.

465-467: What are the i, j subscripts representing? I doubt you want both on both the time and group terms.

465: Given the timing of treatment varies, presumably that also influences treatment success due to weather etc., and you have data available, why not model it as a regression with dummies for the time (or a time trend) rather than put it into 2 period before and after? Please consider.

474: What standard errors are you using given the autocorrelation in the DiD? If you're assuming homoskedasticity, justify or run the model accounting for serial correlation (e.g. cluster robust standard errors or bootstrapping accounting for the autocorrelation). See Bertrand et al. 2004 (<https://doi.org/10.1162/003355304772839588>).

Response to Reviewer Comments:

We thank the reviewers for their thoughtful and thorough comments on our first submission of the manuscript. The document below outlines our specific responses to each comment raised; however, we wanted to highlight several main changes added to the text that may apply to all reviewers' comments:

- 1) Given comments from multiple reviewers, we felt the manuscript could benefit from the addition of a table, outlining the structure of each model included in our analysis, to keep the details of each approach clearer. This is now included in Table 1.
- 2) In the initial text, we did not make it sufficiently clear that there has not been previous documentation about the biophysical characteristics of areas selected for seeding treatments by managers. While there are some recently developed guidelines (i.e. many of the treatments we evaluated predated the guidelines), the guidelines are recommendations and are not mandates. We now discuss in the text many reasons (e.g. the stochastic, unpredictable nature of wildfires, restoration resources, and other factors) why the areas ultimately treated may diverge from these recommendations. Thus, we believe it is valuable to quantify the characteristics of sites where treatments are implemented. We hope we have now discussed this more clearly in the text. In addition, we have investigated several additional biophysical drivers of treatment application, as suggested by the reviewers. Please note that these modifications have resulted in minor changes to the statistics and the sample sizes included in the text, due to refitting of the models (i.e. different number of matched observations, due to matching incorporating a broader set of covariates).
- 3) Reviewer 3 raised an important point about our justification for conducting propensity score matching prior to the DiD models. This prompted us to reconsider elements of our analysis, as we were matching on the basis of time-varying, pre-treatment measures of sagebrush cover, which can hinder inference of the treatment effect, due to the phenomenon referred to as "regression to the mean". Thus, we have revised our approach and conducted DiD on the unmatched dataset (which, given the parallel trends for treated and untreated groups and other met assumptions of DiD, is appropriate). This revised analysis results in a larger positive effect size for seeding treatment, compared to the past submission. We have discussed our revision of this component of the analysis below, under the specific point from Reviewer 3.

Reviewer comments are shown below in *blue italics*. Responses shown in **black**.

In the main text, revisions to the manuscript in response to reviewer comments are shown in blue.

Reviewer #1 (Remarks to the Author):

This paper describes a modeling effort to predict the effect of sagebrush seeding on post-wildfire sagebrush recovery in the western United States. The authors use propensity matching to show that seeded areas have a different bioclimatic/geographic profile than unseeded areas, demonstrating that restoration practitioners deployed seeding with some bias, presumably for

sites where they thought that seeding was more likely to assist sagebrush recovery. A model that ignored this bias suggests that seeding had a negative impact on sagebrush recovery, which is counterintuitive. Using only a subset of samples with similar bioclimatic/geographic profiles, the same model suggested that restoration had no effect on sagebrush recovery. A different type of model (DiD) that accounted for change in sagebrush cover pre-treatment to post-treatment suggested that seeding had a small positive effect on sagebrush recovery. The authors use this final model (actually one of two DiD models with and without a random intercept for fire identity) to demonstrate where seeding had the greatest impact on increasing sagebrush cover.

This paper is one of very few that addresses the issue of selection bias in ecological restoration head on. Moreover, it does so at a spatial scale relevant to the large-scale restoration that is needed to move the needle on biodiversity conservation, climate change mitigation, preventing desertification, and provisioning other ecosystem services. The modeling approach seems like it could (and should) be used in a variety of other ecosystem types to evaluate the true impact of large-scale restoration activities. Such evaluations will be increasingly important over the next decade as restoration funders seek information on the efficacy of their investments, particularly for carbon storage. As such, this well-written paper merits a high-visibility platform.

Response: Thank you so much for your thoughtful feedback.

Major Comments:

1. This study's biggest weakness is that the magnitude of difference in sagebrush cover between seeded and matched unseeded sites was small, +1.1% on average (ranging up to +8.1% in some contexts). This relatively small difference makes it hard to draw a strong conclusion about the efficacy of seeding, particularly when the average difference is exceeded by the measurement error in the RCMAP dataset (6%). Is +1.1% meaningful? Is it worth the expenditure (>\$100M USD since 1990)? The authors are wise to word their conclusions conservatively, without making a clear policy recommendation about sagebrush seeding in the western US, other than to say that the net effect is positive. I do not have a recommendation for addressing this comment other than perhaps to focus more of the discussion on the implications of this work beyond the study system. In my view, the greater importance of this paper is in its novel methodological approach rather than the implications for U.S. restoration policy regarding sagebrush.

Response: Thank you for this comment. We agree that our main goal is to emphasize the novel methodological approach, rather than the implications for sagebrush restoration planning or policy directly (and this was the tone we were hoping to strike in our first submission). Coarse-scale remotely sensed data comes with its own set of clear inferential tradeoffs (including the measurement error mentioned above), so any quantifications of “success” will require further investigation with field studies. However, we aim for this analysis to provide a strong proof-of-concept, illustrating how selection biases may shape the conclusions drawn in studies of restoration and how analytical approaches may be better tailored to account for those biases. We have edited the discussion (and other places in the manuscript) to more clearly

communicate this primary goal of the paper and clarify what next steps (using field studies) would be required to validate any of these results. Overall, we hope that we communicate (in terms of management recommendations) that a) perceptions of “success” of sagebrush management actions may depend both on statistical approach and the characteristics of the sites where restoration trials have been conducted, and b) considering the massive amount of resources being allocated to sagebrush restoration, managers and policy makers should take into account this idea that the “gains” achieved by seeding may vastly differ across sites.

In addition, one of the comments from Reviewer 3 prompted us to revise our use of PSM prior to DiD analysis (see discussion of impacts of “regression to the mean” in DiD, in our response to reviewer 3 below). It appears that our use of PSM prior to DiD may have caused underestimation of the treatment effect (see discussion of regression to mean below). The revised seeding-treatment effect using adjusted statistics is a ~5% increase in sagebrush cover across the study region. While this effect is still small, it is perhaps more worthy of discussion on its own, though we have still worded our conclusions somewhat conservatively, given our use of remotely sensed data.

2. A second weakness, which the authors describe in the discussion, is that some potentially important predictors were not included in the propensity matching and subsequent modeling. The most striking omissions are substrate (e.g., soil type), land use (e.g., grazed versus not), and within-year environmental conditions (e.g., spring soil moisture in the year of seeding). Clearly not every possible variable can be included in this modeling effort, and some variables may simply not be available, but what is the rationale for excluding these particular variables? I think that the effect of within-year environmental conditions on sagebrush seeding success would be especially interesting to restoration practitioners in the sagebrush system.

Response: We have now included variables for soil percent clay and soil percent sand. We opted to include these instead of soil type to make observations across this broad region more directly comparable for the matching process (i.e. using continuous variables, rather than numerous categorical variables across the broad study region) and because past studies have detected an effect of soil percent clay on sagebrush recovery post-fire (see citation in text).

For the effect of within-year environmental conditions, we agree that annual “weather” covariates (distinct from long-term climate variables) related to the year in which restoration occurred would likely have important effects on treatment “success”; however, we did not include them in our analysis primarily because annual weather variables are strongly correlated with climate averages – thus, it is difficult to isolate the effects of short-term weather, compared to long-term climate from single observations made several years after the treatment occurs (in this case, 10-years post-fire). Further, we found that most studies that were attempting to control for the differences between sites used a climate average, so use of a long-term climate variable better reflected the design of the existing studies of sagebrush recovery. Our goal was not necessarily to capture every possible “measurable” source of bias, but rather to construct a model that reflected the types of variables commonly used in

observational studies of sagebrush recovery. Instead, we opted to incorporate the random effect for fire identity, which should account for some of the same temporal variation (as well as variation in space) and was more commonly incorporated into other studies.

RE: grazing, we would love to consider grazing effects but we are not aware of any digitized datasets that accurately summarize grazing history across this broad of a spatial and temporal extent (coauthor Germino works closely with the BLM national rangeland program). As an example of how grazing analyses must typically be done: Williamson et al. (2019, *Biological Invasions*) conducted an analysis of the impacts of grazing on cheatgrass invasion over a 14-year period in four mountain ranges in Nevada and ultimately had to rely on a relatively coarse descriptor of grazing history that was developed through personal communications with USFS managers. They used a variable for whether or not a particular allotment was active and assumed that all active allotments were grazed (while citing the fact that this may not fully capture how and how many cattle were ultimately managed within that allotment). The USGS/BLM Land Treatment Digital Library contains some limited information about grazing within treated areas, but we do not have matching information for untreated sites. Thus, we felt that grazing history was not feasible to accurately summarize across the Great Basin from 1985-2015.

Minor Comments

3. Line 124. I would have benefited from a more thorough description early on of DiD modeling, which I am unfamiliar with.

Response: We have edited the text to include an additional description of DiD in the introduction (highlighted in blue).

4. Line 289. If I understand correctly from the methods, DiD modeling is a clever way of specifying and interpreting a GLM or GLMER to assess change in the response variable (via the treatment \times time interaction) while still ultimately predicting the response variable's state. Wouldn't it be simpler and accomplish the same thing to model sagebrush cover's change over time in treated vs. untreated areas instead? If not, why?

Response:

Difference-in-differences is modeling how the change in sagebrush cover over time differs between treated and untreated areas. The two “differences” in DiD refers to the difference in how much each group changes (i.e. the difference between time=0 and time=1) with treatment. For us, a primary reason for modeling the raw cover value at each time point, rather than the net change, is that it allows us to examine the total amount of sagebrush cover and interpret the treatment effect relative to the “expected” levels of recovery. One of other primary reasons that we adopted that DiD is that the structure for these models is well-defined in the causal inference literature, and we felt this made it clearer what our assumptions were.

It also should be noted that there is more than one model structure that could estimate a treatment effect while considering persistent differences between treated and untreated groups. However, observational studies of restoration success in this system have not commonly (to our knowledge of the literature) modeled change over time in treated v. untreated areas using the method you describe or through DiD (namely because observations of initial sagebrush cover are rarer in the observational datasets).

5. Paragraph starting at Line 379. Could you provide a little bit more information about the RCMAP? How is this product created?

Response: We have now included additional information about RCMAP in a supplementary document (Supplementary Information 3).

6. Line 388. For someone outside of the sagebrush system, it's not clear how 10 years is a relevant time period for sagebrush life history. Is this how long sagebrush seeds remain viable in the soil seedbank? A citation would be helpful.

Response: We hope we have made this clearer in the revised text. The selection of a 10 year time point was to make sure that the stand was beyond the point at which the remotely sensed signal for cover was likely to fluctuate strongly (previously identified as ~ 6 years after fire in Applestein & Germino 2020). This is also an approximate point at which growth rates are less variable (i.e. lower probability of strong declines) based on work presented in Shriver et al. 2019 (Ecol Letters, cited in text). We also found that our results did not qualitatively differ if we examined longer (15-year) or shorter (8-year) timepoints (Supplementary Information 2).

7. Line 393. Why is "sagebrush cover forced to be the same or greater than the previous year in burned areas?" I don't understand why this is necessary. Can't sagebrush cover decline over time too?

Response: This is a data processing action made by the producers of the RCMAP product. The developers of RCMAP ensure that "successive years of estimates in burned areas have values greater than or equal to the previous year" to avoid anomalous decreases associated with low sagebrush-detection at low values of cover and higher levels of background "noise" attributed to bare soils (Rigge, Shi, Homer, Danielson, & Granneman, 2019), though this only applies to the initial years following fire. More detail can be found in the citations described in Supplementary Information 3.

8. Line 463 (equation). "Treatment" and "Group" seem to be used interchangeably between this equation and the preceding paragraph. It would be better to use one throughout.

Response: Thank you for this comment. We now use "group" to more consistently refer to the groups of pixels that either received treatment or did not, which we hope reduces confusion

between the group indicator variable and the treatment effect indicator (interaction between time and group) in the DiD analysis.

9. Figure 5. What are the units on the y axis? The trend is clear, but it's not clear what the magnitude or importance of the pattern is.

Response: The units are percent cover for sagebrush. We have clarified by adding the word “percent” to the axis label.

10. Figure S1. The legend says “post-fire” but the caption and the x-axis label both imply that these trends are supposed to be pre-fire.

Response: We intended for the labels to indicate that these were the pre-fire trends (in sagebrush cover) for pixels that later (following fire) became treated or were left untreated (Untreated post-fire and Treated post-fire). We’ve edited the caption text to try to make this clearer.

Reviewer #2 (Remarks to the Author):

Selection bias in restoration application is an issue when observation studies attempt to draw conclusion from restoration efforts. The idea that selection bias exists in restoration practices is not novel as restoration is not applied at random. The implications of this study are very important and identify some pitfalls with using observations studies to evaluate restoration and other natural resource management practices. The manuscript could be strengthened by including relevant literature on successful sagebrush seedings that contrast the observation studies results to provide a more complete framework for the study and place the results of this study in context with the literature. The study is well-design, sound, and of importance to the field of restoration and land management. The conclusions and interpretations are supported by the data.

1) Ln 22-23 this is not unexpected. Restoration would be more likely needed in more stressful and degraded sites. This could be worded into a hypothesis of the study.

Response: Thank you for this comment. We agree that this is not unexpected and have rephrased our discussion in the introduction to make the novelty of this result a little clearer. Even though it is logical that restoration would likely occur in more stressful or degraded sites, we were trying to communicate that: a) the mechanisms that ultimately determine nonrandom location of restoration treatments have not been quantified in the past; b) the statistical impacts of these nonrandom locations are typically not incorporated into experimental designs or analyses, which could have implication for the inference of treatment effects.

2) Ln 85 tree invasion has also been a major contributor to the decline of Artemisia species in western North America

Response: This has now been included in this sentence.

3) Ln 92-106 The authors need to include research projects (Davies and Bates 2017; Davies et al. 2018, 2019; Urza et al. 2019) that have reported successful sagebrush restoration. This would better frame their argument that observation studies of restoration efforts may have selection bias as well as give a more complete picture of sagebrush restoration variability.

Davies et al. 2018. Rangeland Ecology & Management 71:19-24.

Davies and Bates. 2017. Restoration Ecology 25:33-41.

Davies et al. 2019. Restoration Ecology 27:120-127.

Urza et al. 2019. Biological Invasions 21:1993-2007.

Response: Thank you for these suggestions. The intent of our sentence here was to summarize observational studies that have examined the effects of seeding efficacy over very broad spatial or temporal extents (which have found limited evidence of seeding efficacy), as we were interested in whether sagebrush seeding had been broadly successful. We did not mean to suggest that some experimental studies and studies examining more specific ecological regions within the broader range of sagebrushes have not found positive effects. These are very valuable too, but may be less prone to some of the same issues associated with nonrandom treatment deployment, as they often examine a more constrained set of conditions, years, or vegetation structures. We have clarified this in the text and have now incorporated several of these studies.

4. Ln 110-112 This hypothesis is rather obvious. Undoubtedly restoration treatments are not applied at random. It would defy logic to apply restoration treatments at random. Restoration treatments are applied to areas that will in all likelihood will not recovery on their own. When climate and biophysical traits are more favorable, these sites are allowed to naturally recover. Possibly modify the hypothesis to be more specific or remove it.

Response: We have rephrased this hypothesis to make the novelty of the question a little clearer. While it is obvious that restoration treatments are not applied at random, the primary drivers of their deployment have not been analyzed.

Ln 156 Northern what?

Response: We have now clarified (Northern Basin) in the text.

Ln 207 The authors need to place their results in context with several recent studies that have found higher success with seeding sagebrush than older studies.

Response: Please see above comment

Ln 225-227 cite literature showing that sagebrush natural recovery is greater in the cooler and wetter sites

Response: We have now included the citations in text.

Ln 241 “threatened” is a legal term that I don’t think applies to greater sage-grouse at this time.

Response: This is a typo – thank you so much! We intended to say “candidate.”

Ln 253-254 Need to place a caveat at this point that this is widely known (i.e. not a surprise to anyone that is familiar with restoration practices in drylands).

Response: We have revised the text to make this point clearer.

Ln 262 what about the studies that have detected an increase in sagebrush cover with restoration?

Response: Please see above comment.

Ln 274 existing vegetation also influences soil water. Cheatgrass depletes soil water earlier than native vegetation

Response: Thank you! This is an excellent point, and we’ve integrated it here.

Ln 293 could minor losses from restoration be attributed to seeding of other species (perennial grasses)?

Response: The revised analysis (see response to Reviewer 3) shows fewer instances of minor losses, so we have not integrated this here, but we think this is an interesting possible mechanism.

Ln 353 use citation number compared to name and year

Response: We have addressed this comment in the text.

Ln 354-355 combustion of herb understories and survival of shrub overstories rarely, possibly almost never, occurs in sagebrush communities. Herbs act as ladder fuel to engage highly combustible sagebrush

Response: We reworded this sentence, as our intent was not conveyed as written. We deleted “...combustion of herb understories and survival of shrub overstories, to...”. Our original intent was to communicate that we and others have observed patchy shrub survival where understories of short-statured herbs rapidly pass low flame lengths across the landscape, also leaving partially combusted litter layers. This is common in fires that occur in earlier season, when deep-rooted perennials still have some water but annual grass canopies are dry and combustible. Some land managers refer to these as “dirty burns” and are undesirable conditions because a clean/bare soil template is not available for restoration seedings or herbicide applications (see photos and description in Germino et al. 2016).

Ln 355 and 358 include citations

Response: Thank you! We have now included citations for these ideas.

Ln 558 citation format is different than other citations

Response: Thank you! We have fixed this in the revision.

Ln 599 citation is lacking journal, volume, and page number

Response: Thank you! We have fixed this in the revision.

Reviewer #3 (Remarks to the Author):

Comment: *Simler-Williamson & Germino revisit the benefits of post-fire re-seeding efforts on sagebrush recovery with a particular focus on the potential for selection bias to confound statistical estimation. They compare a naïve statistical model with models that address selection on observables (propensity-score matching) and selection on unobservables (difference in difference). It is an interesting application of causal inference methods. However, I have several important methodological concerns related to all of the models the authors present.*

Response: Thank you so much for the thoughtful comments about the methods below. In particular, your comment about conducting DiD on the matched subsample eventually prompted a more significant change to the analysis (though the general conclusions remained the same), while all other modifications and clarifications were more minor. We believe that each of these changes has improved the manuscript.

Comment: *1) With regard to the naïve statistical model, no details are provided regarding the model specification. For example, what biophysical covariates were included, did they account for the year of the fire, the ecoregion, how many different species were seeded? Or was this a t-test ignoring everything except treatment? It would be far more useful if the “naïve” model was based on a prior publication so readers could assess how impactful the PSM and DiD approaches are relative to current understanding in the field.*

Response:

We hope that our revisions to the manuscript make our intended logic and our model structures (now summarized in Table 1) much clearer here, compared to our initial submission.

Our “naïve” model was a model that only included an effect for treatment. While we agree that this sort of model structure may be unrealistic, our intent was to illustrate the overall possible bias, if no systematic differences between treated and untreated sites were accounted for, as a conceptual starting place. We intended the “matched” model to include as many of same

biophysical covariates that have been included in past studies as possible (which we now elaborate upon in the text), to provide a means of comparison to the DiD approach.

In addition to the regression on the matched subset, we have now added an additional regression that reflects your recommendation. This model includes biophysical covariates that have been considered in past studies (including ecoregion), with a varying intercept for fire identity, which may account for comparison across broad spatial and temporal extents. We have tried to make it clearer how the variables we have included are connected to other studies in the existing literature with the included citations.

In both cases, (regression containing environmental covariates and regression on the matched subset), the treatment effect is not statistically different than zero.

2) With regard to the propensity score matching, little information is provided regarding balancing diagnostics, how time-varying treatments (ie seeding in different seasons of different years) were considered, or finer details regarding the matching algorithm (was it with replacement or 1:1, if with replacement, how was the presence repeated observations accounted for in the errors). It is not clear if all of the variables influencing the treatment decision are included, and the argument is not explicitly made that they are (the defining assumption of the method).

RESPONSE:

We have now included more detailed information about the PSM methods and the about the impacts of the matching process on balance of the covariates, both in the text and in the supplementary information. According to our tests for differences in means (among treated and treated groups), all examined covariates differed before PSM and are balanced in the matched subset. We have reproduced an abbreviated version of the requested information here to facilitate the review process (in Supplementary Information 1).

Nearest neighbor matching of treated and untreated observations on the basis of their propensity scores was completed using the *MatchIt* package in R (cited in text). We conducted nearest neighbor matching using all biophysical covariates mentioned in the main text (and shown in Figure 2). In *MatchIt*, nearest neighbor matching is conducted by computing a propensity score distance between each treated and each control observation, using logistic regression. Treated observations are then assigned a control match by proceeding through the list of treated units and selecting the closest eligible control unit, based on the propensity score. To ensure that covariates would be balanced between treated and untreated groups, we limited the allowable distance between pairs' propensity scores, using a "caliper width" criteria (which determines the maximum allowable difference between paired sites) equivalent to a quarter of the standard deviation of the mean propensity score, as recommended in Guo and Fraser 2010. Unmatched observations (that exceeded the caliper width) were discarded, restricting the sample to a common region of support.

We conducted tests for differences in the means for each covariate between treated and untreated groups, using linear and generalized linear models (depending on the structure of the tested variable) containing a single categorical variable for treatment group. Differences in means were identified based upon the parameter estimate for treatment group. Groups were considered to have statistically important differences in means if the 95% credible intervals for the treatment group parameter did not contain zero. The results of these difference in mean tests are shown in summary Figure S2 in the Supplementary Information. The full distributions for each variable, before and after PSM, are shown in Figure S3 in the Supplementary Information. In the unmatched sample, treated and untreated locations statistically differed in their biophysical characteristics, but in the matched subsample (following PSM), biophysical variables were statistically similar among treated and untreated locations, based on our difference in means tests.

We discuss considerations related to seasonality of treatments in a related minor comment below.

***Comment:** 3) With regard to the difference-in-difference, the interpretation of the interaction term is not easily translated from a linear model to the nonlinear model used here, and the model does not account for group means in the way it would if the model was linear. This should be addressed. Additionally, what is the methodological justification for using DiD on the PSM matched subsample? If selection bias only affects the pre-period, it wouldn't be an issue with DiD, it would have to affect the trends. Results from the unmatched sample are not included in the results figures, and it is not specified what standard errors the authors use to account for serial correlation of the errors in the DiD.*

Response:

The comment above has three main components, which we address below:

- (a) the interpretation of interactions and identification of the DiD treatment effect in our nonlinear model;
- (b) the methodological justification for using DiD on the PSM matched sample;
- (c) specification of the standard errors to account for serial correlation of the errors in the DiD.

a) The interpretation of interactions and identification of the DiD treatment effect in our nonlinear model:

Thank you -- we should have included more detail about the consideration of the treatment effect in the nonlinear model and more precise of our use of the phrase "treatment effect" throughout the text.

To calculate the treatment effect from our DiD models, in the initial submission, we followed the recommendations outlined in Puhani 2012 (*Economics Letters*, now cited in text), which describes the treatment effect as "the cross difference of the observed outcome minus the

cross difference of the potential non-treatment outcome” and that the interaction term represents “incremental effect” of treatment.

The treatment effect can be calculated as:

$$= \Phi (\alpha + \beta + \gamma + X\theta) - \Phi (\alpha + \beta + X\theta) .$$

Where phi = the nonlinear link function; alpha = the time effect; beta = the group differences; and gamma = the interaction between treatment and time (which, in a linear DiD model would represent the treatment effect). To make it clearer that we had considered the differences in calculating a treatment effect for a nonlinear model (as opposed to a linear DiD model), we have now included more explicit information about this calculation in the text.

Further, in several places in the text, we referred to the interaction between the time period and treatment groups as “the treatment effect,” in part because this parameter indicated the directional effect of the treatment (positive/negative), even if its effect size could not directly be interpreted as the treatment effect itself (due to the link function used). This language is imprecise, given the difference in interpretation between a linear DiD v. nonlinear DiD model that you raise, and we have remedied this by referring to this term specifically as the “interaction between the time indicator and group indicator” or “the interaction term” wherever possible.

We also now present the calculated treatment effects (again, as outlined in Puhani 2012) for all models in a new panel of Figure 4, alongside the untransformed parameter estimates for the term associated with the treatment effect.

b) Reasoning for using DiD on the PSM matched subsample:

We thank you for this important point, and we have revised our analysis in response to this comment. We initially conducted DiD on the PSM subsample because, in our experience, DiD approaches were often used in conjunction with propensity score matching to increase the probability that the two treatment groups will have parallel trends. However, we realize that we had not considered the possible impacts of regression to the mean in our DiD approach, and this comment caused us to reconsider.

Daw and Hatfield (2018) demonstrate that matching observations based on time-varying variables, including pre-treatment outcomes, can bias results due to regression to the mean (i.e. if a selected group contains more extreme values for a given variable, subsequent observations for that variable will tend to be closer to the mean). The use of pre-fire measures of sagebrush cover in our original PSM process possibly biased the time*group interaction term estimate consistent with the issues Daw and Hatfield (2018) raise.

In conclusion, we have now adjusted the DiD to examine the full/unmatched dataset, which we agree is a more appropriate approach that avoids the potential for bias associated with

regression to the mean and ensures treated and untreated groups fulfill the assumption of parallel trends before matching.

The estimate from the “unmatched” DiD is more strongly positive than the previously presented result. We believe this is because treated areas tend to have lower pre-fire sagebrush cover than untreated areas, so we may expect matching to select a group of untreated locations that have lower pre-treatment values of sagebrush cover to better match treated locations (see Figure 2, Figure 3). This may cause untreated pixels to “bounce back” to their mean (toward larger sagebrush cover values in later, post-fire timepoints), biasing treatment effects toward negative values.

Just to be thorough, we now also include a supplementary analysis (Supplementary Information #2) in which we conduct PSM only on the time-invariant covariates described in the main text (i.e. no sagebrush cover variables) before the DiD model. In this approach, the DiD parameter estimates are similar to those now found in the main text (i.e. matching on time-invariant variables before DiD has minimal impacts).

c) Accounting for repeated measures:

Thank you for this important point, given the potential impact of this cluster structure on the size of our standard errors. To account for the repeated observations of individual pixels, we have now included a varying intercept/random intercept for pixel identity (i.e. which pair of before/after measurements an observation belongs to). This varying intercept for pixel identity now appears in both versions of the DiD models. In the model that also incorporates a varying intercept for the identity of fire events, the pixel identity cluster is nested within the fire identity cluster.

Specifying cluster structure in a model directly has been discussed as an appropriate approach for accounting for serial correlation in DiD analysis (Cameron and Miller 2015, now cited in text). We preferred using this approach to a post-estimation method for standard errors (e.g. cluster-robust standard errors) because modeling the correlation directly (using varying intercepts) was more complementary to the existing analysis (which already had one model containing a varying intercept to account for the cluster structure associated with fire identity).

Incorporating this cluster structure into our model does not alter our overall interpretation from the analysis. All figures and statistics in text have been updated accordingly.

Line-by-line comments

Comment: Line 70-76: It would be better to phrase these methods with respect to the problem they address and the key assumptions. One is assuming variables driving selection are all observable and measured, which is a high bar.

Response: We have altered the framing of this paragraph to focus less on restoration specifically and more on the broader statistical problem that these methods address. We have also clarified that PSM cannot account for unobserved, unmeasured drivers of selection.

Comment: *Line 152-157: Is there not a record or protocol of how seeding locations are chosen?*

Response: There is not a strict protocol or set of digitized records about how seeding locations are chosen. There are a series of general guidelines found in the BLM's Emergency Stabilization and Rehabilitation document (2007, cited in text), as well as a more recent set of sagebrush-specific guidelines based on which sites are predicted to be most resilient to wildfire, most resistant to cheatgrass invasion, and most suitable as sage grouse habitat (Chambers et al. 2014, Pyke et al. 2017, Chambers et al. 2017). However, there are many reasons why treatment implementation may depart from these guidelines, including interactions between national-, regional-, and local-level coordination or the stochastic nature of wildfires, social capital, funding availability, and weather (now described in text). Despite the existence of more recent sets of guidelines about site prioritization (which predate many of the observations analyzed here), we still think it is valuable to identify where restoration has been implemented in the past, quantify whether treatments have been successful in these sites, and attempt to understand how the characteristics of these sites may have influenced treatment success (and its inference).

Comment: *Line 161: To support the claim that covariates are similar in the matched subsample, please include a summary stats table (or figure) with biophysical characteristics for treatment and control (separately) in the full sample, and in the PSM sample.*

Response: We have included an additional figure in the supplement (Supplementary Information 1, Figures S2 and S3) illustrating that the summary statistics for the biophysical characteristics of treatment and control groups are similar in the PSM sample. We also previously included Figure 2B to illustrate the effect of the PSM process. Figure 2B shows the effect of each biophysical characteristic on the probability of a location being treated, before and after the matching process. For the unmatched full sample, the 95% credible intervals for the parameters associated with each biophysical variable do not include zero, suggesting that these variables have a nonzero effect on the probability of a location receiving treatment (shown in blue in Figure 2B). Following the matching process, the credible intervals for the effect of each characteristic now included zero (shown in grey in Figure 2B), indicating that in the PSM subsample, these variables no longer had an important effect on the probability of a location receiving treatment.

Comment: *Line 167-171: What was the naïve model? Effectiveness of re-seeding is a common research question, please show the results you'd get using a model from the published literature, and then the improvement with PSM and DiD. Presumably the state of the science*

isn't comparing seeded unseeded without controlling for anything, or if it is, it is comparing sites within the same fire, no?

Response: For this comment, please see our response to major comment #1, which discusses the same concern in more detail. We have added an additional model containing several environmental covariates (based on the existing literature) to address this concern.

Comment: *Line 190-197: Is satellite data across the two decades of this study equally likely to misclassify sagebrush? That might be a tough argument. At least provide evidence if that is the argument. If it does change, it seems like a potential weakness to discuss in the discussion. Also clarify that "repeated measures" only applies to your DiD results.*

Response: This is an important point. Rigge et al. 2019 (see Supplementary Information #3) provides a specific analysis of the spatial and temporal accuracy of the RCMAP product (formerly known as the NLCD "Back-in-time" (BIT) dataset), using high resolution satellite imagery (capable of discriminating sagebrush) spanning an 11-year period. Temporal correlations between the high-resolution satellite imagery and RCMAP data had an average R2 of 0.6. While this validation does not stretch back as far as 1984, this study does provide some evidence about the strong predictive performance of the product over a broad time frame.

We have now discussed this potential weakness as well as this validation study's use of field data over a broad temporal period in Supplementary Information #3.

We also have clarified that repeated measures applies only to our DiD results.

Comment: *Line 217: The parenthetical statement is concerning. Please explain when a site is or isn't recorded.*

Response: In this statement, we were primarily trying to acknowledge that, in any observational dataset, there is some probability (even if small) that an observation may fail to be detected or recorded. However, the LTDL dataset represents a thorough effort to document land treatments, and we believe this is a confusing parenthetical to include. We have removed it from the text, as we have confidence in the LTDL dataset's thoroughness.

Comment: *Line 249-250: Do your empirical results reflect knowledge of the treatment decision process from the BLM plans? Do the BLM plans spell out how the treatment decision is made?*

Response: There is not a strict protocol or set of digitized records about how seeding locations are chosen. There are a series of general guidelines found in the BLM's Emergency Stabilization and Rehabilitation document (2008, cited in text), as well as a more recent set of sagebrush-specific guidelines based on which sites are predicted to be most resilient to wildfire, most

resistant to cheatgrass invasion, and most suitable as sage grouse habitat (Chambers et al. 2014, Pyke et al. 2017, Chambers et al. 2017). However, there are many reasons why treatment implementation may depart from these guidelines, including interactions between national-, regional-, and local-level coordination or the stochastic nature of wildfires, social capital, funding availability, and weather (now described in text). Despite the existence of more recent sets of guidelines about site prioritization, we still think it is valuable to identify where restoration has been implemented in the past, quantify whether treatments have been successful in these sites, and attempt to understand how the characteristics of these sites may have influenced treatment success (and its inference).

***Comment:** Line 253-257: The purpose of this paper is getting confusing. Is it to isolate whether seeding is an effective restoration strategy or is it to understand underlying determinants of seeding by the BLM? If the latter is a part of it, do you learn something from the analysis that isn't outlined in the BLM plans? In other words, it is apparent from the empirical analysis that something besides what the BLM outlines as important determines where seeding occurs?*

Response: We hope that we have clarified this comment in the revised text. We were aiming to address each of these goals, as we think that understanding the efficacy of restoration is dependent on understanding where restoration has actually occurred. We aimed to address: 1) Which biophysical characteristics differ between treated and untreated burned areas? 2) Given this information, how does the statistical consideration of the nonrandom application of restoration influence the size and direction of identified treatment effects?; and 3) Are post-fire seeding treatments improving sagebrush recovery across a broad spatiotemporal extent, given the patterns in their existing implementation and given what can be learned from RCMAP?

While there are recommendations in place for seeding treatments, as discussed above, these frameworks are relatively new (compared to the observations in this study) and are guidelines, not fixed rules – in the main text we now discuss numerous reasons why actual treatments' locations may depart from prioritization recommendations. To our knowledge, there has been no documentation of the biophysical characteristics of where treatments actually are implemented, and no other observational studies have quantified this bias. While many of the variables identified as drivers of treatment occurrence are discussed as conceptual criteria in the BLM documents, we believe it is important to quantify this directly. Others (like distance from roads, pre-fire sagebrush cover, and fire size) are not mentioned in BLM documents but appear to have effects.

***Comment:** Line 266-271: Why wasn't soil quality included in the PSM if it is a known and observable (e.g.SSURGO) drivers of treatment and outcome? The underlying assumption of PSM*

is that all information that can influence the probability of treatment is included (such that conditional on these observable covariates, treatment is as good as randomly assigned).

Response: We have now included soil percent clay and soil percent sand (from a dataset that integrates SSURGO and the STATSGO product (where SSURGO is not available)) as a covariate in the PSM.

Comment: 393: *More info needed on RCMAP data accuracy and methods, at least in the SI.*

Response: We have included a new Supplement (#3) containing this information. R^2 for the sagebrush component of this product is ~ 0.60 , based on a variety of validation efforts.

Comment: 405-407: *Clarify wording*

Response: We have now clarified this wording.

Comment: 407: *Rather than jumping into the empirical methods, first explain the major underlying assumption of PSM and its meaning.*

Response: We have now explained the underlying assumption of PSM and its meaning in this section.

Comment: 414-416: *Is this based on the BLM planning document alluded to in the discussion?*

Response: Yes, the covariates that appear in the PSM reflect both recent frameworks for the prioritization of sagebrush restoration sites, as well as covariates that been included as “control” variables in previously-published models of post-fire recovery of sagebrush and our knowledge of the system. We have tried to make this clearer in the revised text.

Comment: 417: *Collinearity in the propensity score is of much less importance than satisfying the assumptions of conditional independence. You aren't making statistical inference with the propensity score, rather just interested in the predicted values of the model. 0.5 seems overly strict given the lesser importance of collinearity here. Please try other version of the pscore model.*

Response: We agree with this comment – we were blurring our goals here, since we were also interested in understanding some of the drivers of treatment location. The collinear variables were primarily elevation, heatload, and winter climate variables (which were strongly correlated with the spring climate variables). We have revised the PSM to include a wider variety of variables (soil percent clay, soil percent sand, heatload, Nov-January climate, and elevation have been added now). We still think it's valuable to be able to infer and discuss how

different variables are related to the probability of treatment. So we constructed a Bernoulli glm (with treatment occurrence as the response variable) containing a subset of the variables from the PSM model (with correlations <0.6) as an additional component of the analysis, to aid in this discussion.

Comment: 427: *Did covariates balance? Please provide what diagnostics were tested and the results. A summary stats table would be helpful here (the same one as requested in the above comment).*

Response: Yes, all covariates balanced. For more detail, please see the comment about balancing of covariates above and the information now provided in Supplementary Information 1.

Comment: 428-434: *Please provide more details on the matching algorithm. Was it NN with replacement or without. If repeated, did you adjust for repeated observations? Was it optimal or greedy matching to select the nearest neighbor. This info can be in the SI, but should be available.*

Response: This information is now provided in Supplementary information 1.

Comment: 437: *What is the methodological concern for doing DiD on the matched sample rather than the entire sample? Selection would need to affect the trends across time not just the pre-period to be an issue. It isn't clear to me how that is plausibly occurring, please clarify. Please include the DiD on the unmatched sample in the results figure.*

Response: Please see comments about this question under major comment 3. This comment was helpful, and the reviewers were correct that matching on the DiD sample was neither necessary nor desirable.

Comment: 444-447: *The interpretation of the DiD interaction term isn't the same in non-linear models as it is in a linear model. The assumptions required for interpreting the interaction term as the DiD in a nonlinear model negate the utility of using the DiD in the first place and don't reflect the logic used here to justify the use of the DiD (lines 450-457). See Lechner 2010 (<http://dx.doi.org/10.1561/08000000014>). This is an important problem to fix for all of the DiD models.*

Response: We discuss this comment in more detail above, under major comment #3, section A.

Comment: 465-467: *What are the i, j subscripts representing? I doubt you want both on both the time and group terms.*

Response: Thank you for this! This was a typo – initially, we had indexed for a hierarchical model to account for the repeated observations of locations. We believe we have now fixed this issue (and the equations themselves have been moved to a table to improve organization).

Comment: 465: *Given the timing of treatment varies, presumably that also influences treatment success due to weather etc., and you have data available, why not model it as a regression with dummies for the time (or a time trend) rather than put it into 2 period before and after? Please consider.*

Response:

Unfortunately, information about exact timing of treatments is not consistently reported in the LTDL. Often, the seeding dates described are start dates found on seeding contract documents, but often the actual seeding dates can be days to months after then. For this reason, it seemed most appropriate to consider a single “before” period, representing a window between a particular fire and the subsequent growing season. While there could be relevant impacts about the timing of treatments, it is unfortunately outside of the scope of this particular analysis to consider. Tolerating the resulting “noise” is one of the tradeoffs in working with a spatially and temporally extensive dataset. However, we think it would be very interesting to explore in future work whether seeding under certain conditions could increase the positive effect observed in this analysis (based on an average across all seeding timings).

Comment: 474: *What standard errors are you using given the autocorrelation in the DiD? If you're assuming homoskedasticity, justify or run the model accounting for serial correlation (e.g. cluster robust standard errors or bootstrapping accounting for the autocorrelation). See Bertrand et al. 2004 (<https://doi.org/10.1162/003355304772839588>).*

Response: We have discussed this comment in more detail above, under Major Comment #3, Part C.

Thank you!

REVIEWER COMMENTS

Reviewer #1 (Remarks to the Author):

The authors have thoroughly addressed my concerns, and this paper is now stronger than it was in the first iteration. As previously noted, this paper makes an important, general contribution to a field of growing importance - evaluating large-scale restoration policies in an unbiased manner. I have no further comments.

Reviewer #2 (Remarks to the Author):

The manuscript is greatly improved. The authors have addressed all my concerns adequately.

Reviewer #4 (Remarks to the Author):

We were asked to review the revision of the paper (as new reviewers) and pay particular attention to whether Reviewer 3's prior comments, which were excellent, were addressed. Thus, this review assesses the paper and which of Reviewer 3's comments require further revisions in our opinion.

This paper is well written and well motivated. It uses three study designs to demonstrate how different approaches to controlling for observed and unobserved variation between treatment and control groups can lead to different estimates of the effect of a restoration effort (in this case, post-fire reseeding of sagebrush). From a methodological perspective, the work that this paper sets out to do -- demonstrating the value of sophisticated causal inference methods -- is important for the fields of ecology and restoration ecology. The authors do a good job of explaining the potential sources of selection bias and how it can impact estimates of treatment effectiveness.

However, we have several suggestions to strengthen the paper and enhance its methodological contribution:

1. Most importantly, the paper ignores the potential for time-varying confounding variables, which are likely common. Why not use a panel approach? Why are time-varying confounding variables ignored?
2. The authors could strengthen the paper by further assessing why the results/estimates differ, based on the validity of the assumptions of the different approaches -- which ones should we "trust" or make the assumptions that are the most plausible for the data and context?
3. All five of the models presented use negative binomials, but the authors do not provide a justification for this choice or discuss the potential sensitivity of their results to the functional form of the model.
4. We understand that the "naive model" is meant to demonstrate the impact that ignoring sources of bias will have on the estimated treatment effect, but it seems implausible that land managers would use such a simplified model. We suggest that the authors provide information on the types of models that are currently used to estimate the treatment effectiveness, to give the reader a sense of the actual baseline models that the authors are responding to. Thus, we agree with Reviewer 3's comment and feel that it is insufficient to do a "naïve" model ... that only includes an effect for treatment" - that is a strawman that does not exist in the literature because

people performing observational analyses typically do include some control variables. We agree with the need to implement R3's suggestion instead and are repasting that comment: 1) With regard to the naïve statistical model, no details are provided regarding the model specification. For example, what biophysical covariates were included, did they account for the year of the fire, the ecoregion, how many different species were seeded? Or was this a t-test ignoring everything except treatment? It would be far more useful if the "naïve" model was based on a prior publication so readers could assess how impactful the PSM and DiD approaches are relative to current understanding in the field.

5. Why were propensity score matching and diff-in-diff the methods chosen over other approaches (e.g., panel regression)?

6. Panel regression, propensity score matching, and difference-in-difference models have different underlying assumptions that will impact their implementation and the interpretation of the results. We suggest adding some discussion about these differing assumptions to the methods or discussion section of the manuscript. Some potential references to cite are listed below:

Butsic et al. 2017. Quasi-experimental methods enable stronger inferences from observational data in ecology. *Basic & Applied Ecology*, 19.

Larsen et al. 2019. Causal analysis in control-impact observational studies with ecological data. *Methods in Ecology & Evolution*, 10(7): 924-934. (particularly Table 1)

Athey, S. & Imbens, G. *The State of Applied Econometrics - Causality and Policy Evaluation*. 31, 3-32 (2016).

Imbens, G. W. & Wooldridge, J. M. Recent Developments in the Econometrics of Program Evaluation. *J. Econ. Lit.* 47, 5-86 (2009).

Ferraro, P. J. & Hanauer, M. M. Advances in Measuring the Environmental and Social Impacts of Environmental Programs. *Annu. Rev. Environ. Resour.* 39, 495-517 (2014).

Ferraro, P. J. & Miranda, J. J. Panel Data Designs and Estimators as Substitutes for Randomized Controlled Trials in the Evaluation of Public Programs. *J. Assoc. Environ. Resour. Econ.* 4, 281-317 (2017).

Angrist, J. D. & Pischke, J. *Mostly Harmless Econometrics: An Empiricist's Companion*. (Princeton University Press, 2009).

Greenstone, M. & Gayer, T. Quasi-experimental and experimental approaches to environmental economics. *J. Environ. Econ. Manage.* 57, 21-44 (2009).

7. Previous papers using matching approaches in conservation science/resource management have combined pre-regression matching with mixed effects regression or panel regression models to control for both observable and unobservable differences between the treatment and control groups. How would the results of a mixed effects regression model on the matched dataset differ from your difference-in-difference results?

Citations:

Starrs et al. 2018. The impact of land ownership, firefighting, and reserve status on fire probability in California. *Environmental Research Letters* 13(3).

Jones & Lewis. 2015. Estimating the counterfactual impact of conservation programs on land cover outcomes: the role of matching and panel regression techniques. *PLoS One* 10(10).

7. We felt that Reviewer 3's comment about the error term was insufficiently addressed Comment: "Results from the unmatched sample are not included in the results figures, and it is not specified what standard errors the authors use to account for serial correlation of the errors in the DiD." It is still not clear what type of standard errors the authors use, and what they are clustering on. Line 205 does not answer the Reviewer 3's question "beyond the varying intercept included to account for autocorrelation between repeated observation." Do you use clustered robust standard errors given the reference to Cameron and Miller? If so, what do you cluster on?

Editor: Following further consultation with reviewer #3 on your response to comment #3 from the previous review, we would like to request that you include cluster robust standard errors in your Supplementary Information for your main DiD results.

Other comments:

Line 40 - "However, outcomes of restoration treatments can be highly variable – even across similar sites and treatments^{1,2} – and this variability in both space and time^{3,4} can obscure efforts to quantify restoration effectiveness" Couldn't another issue be that it is hard to control for confounding variables in observational data? so the variability could be due to true heterogeneity in the effectiveness, but also bias in prior estimates due to different degrees of controlling for confounding variables?

Line 70 - Why focus only on these methods (eg vs other approaches or quasi experimental ones)? E.g., see

Athey, S. & Imbens, G. The State of Applied Econometrics - Causality and Policy Evaluation. 31, 3–32 (2016).

Larsen et al. 2019. Causal analysis in control-impact observational studies with ecological data. *Methods in Ecology & Evolution*, 10(7): 924-934.

Imbens, G. W. & Wooldridge, J. M. Recent Developments in the Econometrics of Program Evaluation. *J. Econ. Lit.* 47, 5–86 (2009).

Ferraro, P. J. & Hanauer, M. M. Advances in Measuring the Environmental and Social Impacts of Environmental Programs. *Annu. Rev. Environ. Resour.* 39, 495–517 (2014).

Ferraro, P. J. & Miranda, J. J. Panel Data Designs and Estimators as Substitutes for Randomized Controlled Trials in the Evaluation of Public Programs. *J. Assoc. Environ. Resour. Econ.* 4, 281–317 (2017).

Angrist, J. D. & Pischke, J. *Mostly Harmless Econometrics: An Empiricist's Companion*. (Princeton University Press, 2009).

Oster, E. Unobservable Selection and Coefficient Stability: Theory and Evidence. *J. Bus. Econ. Stat.* 0, 1–18 (2017).

88 - How can they "examine the effects of selection bias"? These analyses make assumptions about the degree to which select bias is addressed, and the assumptions are often untestable. I suggest revising this sentence.

Response to Reviewers

Dear Reviewers,

Thank you for the thoughtful questions and critiques below. Before addressing specific comments, we would like to provide an overview of the major changes made to the manuscript, following your recommendations:

- 1) We have now included a within-estimator panel regression, which extends the DiD framework to incorporate multiple time points. This model structure (recommended by Reviewer 4 and discussed in the prior review by Reviewer 3) allowed us to examine observations over the 10 years following treatment, incorporate time-varying covariates (such as weather variables specific to each year), and address time-varying sources of heterogeneity between groups. Given this substantial addition, we have made a few modifications to the existing manuscript:
 - a. We no longer compare results for multilevel models containing varying intercepts for fire identity and site identity to models containing only a varying intercept for site identity. Overall, we believe the former is a more conceptually complete model regardless (accounting for both cluster structures), so we have chosen to omit the latter structure for simplicity. This allowed us more room in the results to present and discuss the new panel regression. The WE panel regression contains the same varying intercepts for consistency.
 - b. Figure 4 now presents the results of predicted trajectories of sagebrush cover predicted by the w-e panel regression, and we have moved the additional parameter estimates from these models (outside of the predicted average treatment effect for treated sites) to the Supplementary material.
 - c. We have modified text in the results, methods, and discussion to present this new analysis.
 - d. Given the addition of the panel approach, our exact data points (which were randomly selected from spatial datasets) have shifted slightly, because part of our selection criteria for points shifted. In the revised analysis, points had to have sagebrush cover data for each of the 10 years following treatment (rather than simply Year 0 and Year 10). Missing values are possible in the remotely sensed RCMAP product due to a variety of factors related to the remote sensing of plant cover, so this new criteria shifted our exact datapoints. Consequently, the results of the models marginally differ (all analyses have been re-run), but the overall interpretation is the same. In the results section, updates to exact statistics have been highlighted in purple text.

- 2) In the supplementary information, we have now included results from a comparative DiD and panel regression analysis that calculates cluster-robust standard errors, rather than using a multilevel structure to account for repeated measures. We have also strengthened our discussion of how we acknowledged cluster structure in our modeling approaches.

- 3) We have expanded the discussion to better incorporate a comparison of the limitations and assumptions of each statistical approach.

Our complete comments are below, in black text. Reviewer comments are shown in *blue italics*.

REVIEWER COMMENTS

Reviewer #1 (Remarks to the Author):

The authors have thoroughly addressed my concerns, and this paper is now stronger than it was in the first iteration. As previously noted, this paper makes an important, general contribution to a field of growing importance - evaluating large-scale restoration policies in an unbiased manner. I have no further comments.

Response: Thank you for your previous thoughts on the manuscript!

Reviewer #2 (Remarks to the Author):

The manuscript is greatly improved. The authors have addressed all my concerns adequately.

Response: Thank you so much for your past comments!

Reviewer #4 (Remarks to the Author):

We were asked to review the revision of the paper (as new reviewers) and pay particular attention to whether Reviewer 3's prior comments, which were excellent, were addressed. Thus, this review assesses the paper and which of Reviewer 3's comments require further revisions in our opinion.

This paper is well written and well motivated. It uses three study designs to demonstrate how different approaches to controlling for observed and unobserved variation between treatment and control groups can lead to different estimates of the effect of a restoration effort (in this case, post-fire reseeding of sagebrush). From a methodological perspective, the work that this paper sets out to do -- demonstrating the value of sophisticated causal inference methods -- is important for the fields of ecology and restoration ecology. The authors do a good job of explaining the potential sources of selection bias and how it can impact estimates of treatment effectiveness.

However, we have several suggestions to strengthen the paper and enhance its methodological contribution.

Response: Thank you for this very constructive feedback. Below, we have tried to adopt these suggestions or provide clearer justification for our decisions. We believe that considering these comments has strengthened the analysis, and we appreciate the thorough critique.

1. Most importantly, the paper ignores the potential for time-varying confounding variables, which are likely common. Why not use a panel approach? Why are time-varying confounding variables ignored?

Response:

In the revised MS, we have now provided a within-estimator panel regression approach that incorporates possible time-varying variables.

Our primary reason for conducting first differences panel regression/DiD with a single before-after time point, rather than a full panel regression was the large size of our dataset (20,000) – a full panel approach greatly decreases the tractability of fitting our model (over 10 years post-fire).

However, we fully agree that time-varying confounders are of possible concern. Further, time-varying factors are also of significant management interest to restoration practitioners, as cited in the text. In particular, spring weather (i.e. interannual variation in climate) in the years following treatment application has been cited as a key reason why restoration outcomes may diverge from expectations, beyond what is captured by longer-term climate averages (like the ones included in our analysis).

We have now included a full, additional comparative approach: a within-estimator panel regression with fixed effects for time-since-treatment and treatment group, with annual observations for years 0-10 following treatment. We refer to this model as “within-estimator panel regression” instead of other possible terms (e.g. “fixed effects”) to avoid confusion with other statistical approaches commonly used in ecology (this terminology is also adopted in Larson et al. 2019 (cited in text) for similar reasons). The within-estimator panel regression incorporates annual spring weather covariates, which results in a slightly shifted estimated treatment effect, compared to the DiD. While there are other possible time-varying confounders that could influence our treatment effect, we argue that weather is likely to be the most important (and most of concern to managers), given the factors influencing treatment designation in this dataset.

In response to this addition, we have made several modifications to the presentation of our results and accompanying figures (as described above). The treatment effect shifts with the inclusion of the full panel approach, compared to DiD, but the models suggest similar conclusions (a moderately-sized positive effect of seeding on sagebrush cover).

2. The authors could strengthen the paper by further assessing why the results/estimates differ, based on the validity of the assumptions of the different approaches -- which ones should we "trust" or make the assumptions that are the most plausible for the data and context?

Response: Thank you for this suggestion. We have now incorporated more information about the assumptions (and limitations) of each approach and provided interpretation of the results in light of these assumptions. These can be found throughout the text, in the introduction, methods, and discussion sections.

3. All five of the models presented use negative binomials, but the authors do not provide a justification for this choice or discuss the potential sensitivity of their results to the functional form of the model.

Response: We have included clearer language in the methods about our justification for the choice of negative binomial glms and its implications (lines ~568).

The sagebrush cover data from RCMAP (our response variable) are positively constrained integers, making the negative binomial distribution an appropriate choice that reflects the data generating process. Traditional OLS linear regression approaches would be inappropriate to apply here, given that the data are discrete & positively constrained. We could have considered applying a Beta distribution here, given that the data can also be interpreted as % sagebrush cover (constrained between 0 and 1); however, Beta distributions cannot include true "0" values, which would require us to either employ an offset or disregard 0s (both of which seemed like stronger assumptions). Therefore, even though negative binomial distributions are not constrained at the upper bound of our dataset, we thought Beta distributions were a less appropriate choice, given the biological importance of 0% cover in this dataset. We also could have considered applying a Poisson distribution (also appropriate for positively constrained count data, under the assumption that the dispersion in the data is equal to the mean occurrence parameter); however, given the very large sample size in our data, we opted to employ the negative binomial distribution because it allows estimation of a separate dispersion parameter to accommodate deviations from the assumption that the mean rate is equivalent to the dispersion. In situations in which data are over or underdispersed, this makes the negative binomial a better choice, and in situations where dispersion is approximately equal to the mean (λ), then the negative binomial and Poisson forms of glms should result in similar results regardless.

Graphical posterior-predictive checks (Supplement 1, Figure S4) indicated that the predictions generated from the model reproduced the overall pattern in the raw data, suggesting that a negative binomial model is an appropriate choice, in terms of model fit. In particular, there was an exceedingly low probability of the negative binomial models predicting values greater than 100% cover.

In the preceding set of reviews, Reviewer 3 asked how we had considered the implications of a nonlinear (negative binomial) DiD, especially in how we calculated the ATE (*With regard to the difference-in-difference, the interpretation of the interaction term is not easily translated from a linear model to the nonlinear model used here, and the model does not account for group means in the way it would if the model was linear. This should be addressed.*)

To take into account the impact of our model's functional form on the calculation and interpretation of the treatment effect, we followed the recommendations outlined in Puhani 2012 (*Economics Letters*, cited in text), which describes the treatment effect from a nonlinear DiD as "the cross difference of the observed outcome minus the cross difference of the potential non-treatment outcome" and that the interaction term represents "incremental effect" of treatment, or:

$$= \Phi (\alpha + \beta + \gamma + X\theta) - \Phi (\alpha + \beta + X\theta) .$$

Where phi = the nonlinear link function; alpha = the time effect; beta = the group differences; and gamma = the interaction between treatment and time (which, in a linear DiD model would represent the treatment effect).

In the revised text, we have included clearer explanation that the negative binomial glms appear to be an appropriate choice based both on the structure of the data/data generating process and the graphical posterior predictive checks for model fit.

4. We understand that the "naive model" is meant to demonstrate the impact that ignoring sources of bias will have on the estimated treatment effect, but it seems implausible that land managers would use such a simplified model. We suggest that the authors provide information on the types of models that are currently used to estimate the treatment effectiveness, to give the reader a sense of the actual baseline models that the authors are responding to. Thus, we agree with Reviewer 3's comment and feel that it is insufficient to do a "naïve" model ... that only includes an effect for treatment" – that is a strawman that does not exist in the literature because people performing observational analyses typically do include some control variables. We agree with the need to implement R3's suggestion instead and are repasting that comment: 1) With regard to the naïve statistical model, no details are provided regarding the model specification. For example, what biophysical covariates were included, did they account for the year of the fire, the ecoregion, how many different species were seeded? Or was this a t-test ignoring everything except treatment? It would be far more useful if the "naïve" model was based on a prior publication so readers could assess how impactful the PSM and DiD approaches are relative to current understanding in the field.

Response:

This is a very helpful comment, and we have tried to put each of the models that we used in better context, in terms of the existing literature and their biological "realism". We do not want the "naïve" model to appear as a strawman, and it was not our intent to indicate that the

“naïve” model was a management-relevant or particularly commonly used model. Instead, the purpose of including this model is to indicate the general direction of the differences between treated and untreated sites (i.e. difference in mean sagebrush cover, 10 years after fire, in each type of site), as a sort of conceptual “starting place” for understanding why estimated treatment effects may differ across statistical approaches. We believe it is important to display, somewhere in the manuscript, a simple difference-in-means to make it clear that untreated sites do, in fact, have systematically greater post-fire sagebrush cover. We believe this conceptual counterpoint makes the mechanisms determining the shifts in effect size with matching, measured covariates, and consideration of unmeasured sources of bias much clearer.

To make this purpose clearer, we have now described this “naïve” analysis as our “null” model, for the purposes of comparison in the text. In this case, the null model captures the treatment effect if measured and unmeasured differences in treated and untreated sites are assumed to be inconsequential.

Based on the existing literature, we believe that the most commonly applied models more commonly resemble the “matched” model and the “control variable” model in our analysis. The covariates included in our PSM and in the “control variable” model each were selected based on variables considered in previous studies, cited in the text. While BACI designs are commonly discussed in restoration, we found fewer examples of BACI approaches in the post-fire restoration literature, and none that employ a BACI design over a broad area. In the revised text, we have now more thoroughly discussed where models containing similar covariates have been applied in the literature, and which covariates they have considered.

(*Inspired by this comment, we also changed the precipitation variable included in our PSM and environmental covariate models to be “Nov-Apr” precipitation, rather than Feb-Apr precipitation, to better reflect covariates used in a wider range of studies of sagebrush restoration).

5. Why were propensity score matching and diff-in-diff the methods chosen over other approaches (e.g., panel regression)?

Response: We believe we have addressed the panel regression portion of this question above, under related Question #1. We have now incorporated within-estimator panel regression.

Regarding the other quasi-experimental approaches that we are aware of (beyond matching & DiD/panel regression):

We did not apply an instrumental variable approach primarily because we lacked a theoretical basis for any IV that was correlated with treatment but uncorrelated with sagebrush cover. As we mention in the text, there has been very little research into the social, economic, and biological factors determining where restoration actions are applied (despite the relative strength of related research about the application of other kinds of conservation actions), so we didn’t feel we could make an argument about a valid instrument.

We did not adopt regression discontinuity methods here because we did not have any evidence that treatment assignment probability exhibited threshold-like behavior, associated with some focal variable.

6. Panel regression, propensity score matching, and difference-in-difference models have different underlying assumptions that will impact their implementation and the interpretation of the results. We suggest adding some discussion about these differing assumptions to the methods or discussion section of the manuscript. Some potential references to cite are listed below:

Butsic et al. 2017. Quasi-experimental methods enable stronger inferences from observational data in ecology. Basic & Applied Ecology, 19.

Larsen et al. 2019. Causal analysis in control-impact observational studies with ecological data. Methods in Ecology & Evolution, 10(7): 924-934. (particularly Table 1)

Athey, S. & Imbens, G. The State of Applied Econometrics - Causality and Policy Evaluation. 31, 3–32 (2016).

Imbens, G. W. & Wooldridge, J. M. Recent Developments in the Econometrics of Program Evaluation. J. Econ. Lit. 47, 5–86 (2009).

Ferraro, P. J. & Hanauer, M. M. Advances in Measuring the Environmental and Social Impacts of Environmental Programs. Annu. Rev. Environ. Resour. 39, 495–517 (2014).

Ferraro, P. J. & Miranda, J. J. Panel Data Designs and Estimators as Substitutes for Randomized Controlled Trials in the Evaluation of Public Programs. J. Assoc. Environ. Resour. Econ. 4, 281–317 (2017).

Angrist, J. D. & Pischke, J. Mostly Harmless Econometrics: An Empiricist's Companion. (Princeton University Press, 2009).

Greenstone, M. & Gayer, T. Quasi-experimental and experimental approaches to environmental economics. J. Environ. Econ. Manage. 57, 21–44 (2009).

Response: Thank you so much for this suggestion. We have moved some of this information from the methods section and added text to the discussion to make clearer comparisons of the assumptions of each analysis approach. We have also incorporated several of these references.

7. Previous papers using matching approaches in conservation science/resource management have combined pre-regression matching with mixed effects regression or panel regression models to control for both observable and unobservable differences between the treatment and control groups. How would the results of a mixed effects regression model on the matched dataset differ from your difference-in-difference results?

Citations:

Starrs et al. 2018. The impact of land ownership, firefighting, and reserve status on fire probability in California. Environmental Research Letters 13(3).

Jones & Lewis. 2015. Estimating the counterfactual impact of conservation programs on land cover outcomes: the role of matching and panel regression techniques. PLoS One 10(10).

Response:

Thank you for this question. We had used this approach (PSM prior to DiD/panel regression) in the first submission of this manuscript, for the reasons you list above, as we were familiar with several studies that had employed those approaches together. However, in the previous set of reviews, Reviewer 3 questioned whether using that approach was redundant, given that the treated and untreated locations exhibited parallel trends before matching.

In response to this set of previous comments, we adjusted the DiD to examine the full/unmatched dataset. Considering the treated and untreated groups exhibited parallel trends pre-matching, we thought this was an appropriate use of DiD. We moved the results from the analysis in which we conduct PSM (on only the time-invariant covariates described in the main text) before the DiD model into Supplementary Information #2. We did this mostly to simplify the presented results, considering that the DiD parameter estimates from the PSM subset are similar to those now found in the main text. However, we are happy to rearrange the results and present the PSM+DiD model results in the main text if desired.

7. We felt that Reviewer 3's comment about the error term was insufficiently addressed

Comment: "Results from the unmatched sample are not included in the results figures, and it is not specified what standard errors the authors use to account for serial correlation of the errors in the DiD." It is still not clear what type of standard errors the authors use, and what they are clustering on. Line 205 does not answer the Reviewer 3's question "beyond the varying intercept included to account for autocorrelation between repeated observation." Do you use clustered robust standard errors given the reference to Cameron and Miller? If so, what do you cluster on?

Response:

We apologize for the confusion here – because the Methods section is at the end of the article, we believe that we did not make our past work to address Reviewer 3's previous comment as clear as we could have.

Failure to consider clustering of observations can result in artificially small confidence intervals. To account for the correlation of observations in our dataset, we opted to model the correlation structure directly as part of the model structure, using hierarchical/multilevel models (also known as random effects or mixed effects models), rather than apply cluster-robust standard errors (CRSE) as a post-estimation strategy. We identified two main clustering structures within our dataset: firstly, we had repeated measures (before and after treatment) for individual locations (n=20,000 locations total, with 2 or 11 observations each, depending on the model), and secondly, our locations were clustered within particular fire events (n= ~1500 individual fires), which could generate both spatial and temporal (with the year of the fire) autocorrelation. To account for this clustering, we included a varying/random intercept for the

identity of each location (given that each location had repeated measures, before and after treatment), nested within a varying/random intercept for the identity of the fire impacting the site, to account for the possible clustering by fire event. Given our knowledge of the system and our model's hierarchical structure, we believe that the primary clustering factors in the model were correctly specified.

We cited Cameron and Miller because they describe hierarchical/multilevel models as a possible alternate approach to CRSE for accounting for clustered errors: *“One way to control for clustered errors in a linear regression model is to additionally specify a model for the within-cluster error correlation, consistently estimate the parameters of this error correlation model, and then estimate the original model by feasible generalized least squares (FGLS) rather than ordinary least squares (OLS). Examples include random effects estimators and, more generally, random coefficient and hierarchical models.”* To our knowledge, CRSE and random effects models have both been described as appropriate ways of accounting for clustering of observations. Wooldridge (2009, *Econometric analysis of cross section and panel data*) also mentions hierarchical/multilevel models as an approach for accounting for cluster structure. In a recent paper, Oshchepkov and Shirokanova (2022, *Bridging the gap between multilevel modeling and economic methods*) extensively review past investigations that compare efficacy of CRSE v. multilevel models in accounting for clustering (including in panel regression). In general, the authors do not describe one approach as clearly superior to the other overall and suggest that multilevel models could be more widely employed in panel regression and other econometrics approaches.

From our personal perspective, our main reason for employing a multilevel model instead of CRSE is that ecologists are a primary audience for our analysis. We have not commonly encountered CRSE in the ecological literature, whereas multilevel/random effects models are frequently used and taught in ecology as a way of ensuring that standard errors and mean estimates reflect clustering of observations.

That said, we definitely want to quell any concerns that the reviewers have about the possibility of artificially small standard errors around our effect sizes, and we would like to demonstrate that our results are robust to our modeling choices. In Supplementary Information 2, we have included an additional version of our models' parameter estimates with cluster-robust standard errors. In the supplementary analysis, we have used a frequentist approach, as we are not aware of any widely-used Bayesian equivalent to CRSE (though Szio et al. 2011 present a Bayesian equivalent to a sandwich estimator, we have not seen this widely applied and did not want to generate additional concerns about how our methods may differ).

To calculate cluster-robust standard errors, we used the “`sandwich()`” package in R, using the `vcovCL()` option, which obtains multi-way clustered covariance matrix estimators for a variety of model forms, including negative binomial glms. For our DiD and panel regression models, we clustered around fire identity and location, to reflect the same clustering factors included in our multilevel model. All other aspects of the models' structures were the same as presented in the text.

As shown in Figure S2.1 and Tables S2.1-S2.2 in Supplementary Information 2, the cluster robust 95% confidence intervals for the parameter estimates (and related p-values) indicate a significant effect of the treatment occurrence indicator variable. The exact estimates and 95% intervals differ slightly from the multilevel versions of the same models, due to the fact that the multilevel models incorporate the cluster-structure into the varying intercept estimates during model fitting (so part of this variation will be absorbed by the varying intercept itself, rather than in the standard errors for the estimate associated with treatment application), while CRSE adjusts the standard errors post-estimation. **However, overall, the multilevel and CRSE-based results lead to similar ecological conclusions.**

We have included additional text in the methods (Line 233) to better explain why we present a multilevel model approach, specify the structure of our clustering factors, and to indicate to the reader that the cluster-robust estimates result in similar inference.

Given that the mixed modeling approach resulted in very similar parameter estimates and intervals (compared to CRSE), we hope that this sufficiently addresses the concerns raised here, but please let us know if there's anything else we can do to improve this aspect of the manuscript.

Citations:

- Cameron, A. C., & Miller, D. L. (2015). A Practitioner ' s Guide to Cluster-Robust Inference, *50*(2), 317–372.
- Szpiro, Adam A., Kenneth M. Rice, and Thomas Lumley. (2010). "Model-robust regression and a Bayesian "sandwich" estimator." *The Annals of Applied Statistics* 4.4: 2099-2113.
- Oshchepkov, A., & Shirokanova, A. (2022). Bridging the gap between multilevel modeling and economic methods. *Social Science Research*, in press.
<https://doi.org/10.1016/j.ssresearch.2021.102689>
- Wooldridge, Jeffrey, M. (2010). Econometric analysis of cross section and panel data.

Editor:

Following further consultation with reviewer #3 on your response to comment #3 from the previous review, we would like to request that you include cluster robust standard errors in your Supplementary Information for your main DiD results.

Response: Thank you for this comment. Please see our response above, to Reviewer #4's Comment #7. The requested supplementary analysis with CRSE can be found in Supplementary Information 2 and results in similar inference to our multilevel model approach.

Other comments:

Line 40 - "However, outcomes of restoration treatments can be highly variable – even across similar sites and treatments^{1,2} – and this variability in both space and time^{3,4} can obscure efforts to quantify restoration effectiveness" Couldn't another issue be that it is hard to control for confounding variables in observational data? so the variability could be due to true heterogeneity in the effectiveness, but also bias in prior estimates due to different degrees of controlling for confounding variables?

Response: Yes, this is a related component in quantifying restoration efficacy. We meant to indicate that variability would make average efficacy difficult to identify. We have clarified our language in this sentence. We agree that confounders are an additional issue, which we discuss in the following paragraphs.

Line 70 - Why focus only on these methods (eg vs other approaches or quasi experimental ones)? E.g., see

Athey, S. & Imbens, G. The State of Applied Econometrics - Causality and Policy Evaluation. 31, 3–32 (2016).

Larsen et al. 2019. Causal analysis in control-impact observational studies with ecological data. Methods in Ecology & Evolution, 10(7): 924-934.

Imbens, G. W. & Wooldridge, J. M. Recent Developments in the Econometrics of Program Evaluation. J. Econ. Lit. 47, 5–86 (2009).

Ferraro, P. J. & Hanauer, M. M. Advances in Measuring the Environmental and Social Impacts of Environmental Programs. Annu. Rev. Environ. Resour. 39, 495–517 (2014).

Ferraro, P. J. & Miranda, J. J. Panel Data Designs and Estimators as Substitutes for Randomized Controlled Trials in the Evaluation of Public Programs. J. Assoc. Environ. Resour. Econ. 4, 281–317 (2017).

Angrist, J. D. & Pischke, J. Mostly Harmless Econometrics: An Empiricist's Companion. (Princeton University Press, 2009).

Oster, E. Unobservable Selection and Coefficient Stability: Theory and Evidence. J. Bus. Econ. Stat. 0, 1–18 (2017).

Response: We have described our rationale above, in response to Question #5, and have now added a panel regression approach.

88 - How can they "examine the effects of selection bias"? These analyses make assumptions about the degree to which select bias is addressed, and the assumptions are often untestable. I suggest revising this sentence.

Response: Thank you! This was imprecise wording, and we have revised the sentence accordingly.

REVIEWERS' COMMENTS

Reviewer #5 (Remarks to the Author):

The authors have thoroughly and thoughtfully addressed our concerns and suggestions. We appreciate the time and care they put into their revisions, and we believe that this manuscript will make an important contribution to the fields of restoration ecology and the application of impact evaluation/causal inference to ecology.

Response to Reviewers

Thank you to the reviewers for the thoughtful and constructive feedback throughout the revision process. Given that the only comment from this most recent submission (below) does not recommend any changes, we have limited our edits to changes requested by the Editorial team, regarding formatting, typos, and other journal-related requirements.

We appreciate your input on this research!

Reviewer #5 (Remarks to the Author):

The authors have thoroughly and thoughtfully addressed our concerns and suggestions. We appreciate the time and care they put into their revisions, and we believe that this manuscript will make an important contribution to the fields of restoration ecology and the application of impact evaluation/causal inference to ecology.